# Stability of neocortical synapses across sleep and wake states during the critical period in rats

**Brian A Cary, Gina G Turrigiano***

Department of Biology, Brandeis University, Waltham, United States

**Abstract** Sleep is important for brain plasticity, but its exact function remains mysterious. An influential but controversial idea is that a crucial function of sleep is to drive widespread downscaling of excitatory synaptic strengths. Here, we used real-time sleep classification, ex vivo measurements of postsynaptic strength, and in vivo optogenetic monitoring of thalamocortical synaptic efficacy to ask whether sleep and wake states can constitutively drive changes in synaptic strength within the neocortex of juvenile rats. We found that miniature excitatory postsynaptic current amplitudes onto L4 and L2/3 pyramidal neurons were stable across sleep- and wake-dense epochs in both primary visual (V1) and prefrontal cortex (PFC). Further, chronic monitoring of thalamocortical synaptic efficacy in V1 of freely behaving animals revealed stable responses across even prolonged periods of natural sleep and wake. Together, these data demonstrate that sleep does not drive widespread downscaling of synaptic strengths during the highly plastic critical period in juvenile animals. Whether this remarkable stability across sleep and wake generalizes to the fully mature nervous system remains to be seen.

***For correspondence:**
turrigiano@brandeis.edu

**Competing interests:** The authors declare that no competing interests exist.

## Introduction

Sleep is a widely expressed behavior, present in animals as evolutionarily distant as Cassiopea jellyfish and the Australian dragon lizard (*Nath et al., 2017*; *Shein-Idelson et al., 2016*). Despite its ubiquity and long history of scientific study, sleep – and more broadly the function of brain states – remains deeply mysterious. Sleep disruption can perturb learning and memory consolidation (*Walker and Stickgold, 2004*; *Krause et al., 2017*), presumably through the modulation of synaptic plasticity (*Frank and Cantera, 2014*). However, there is little agreement on the nature of this regulation. Researchers have variously proposed that sleep stabilizes, strengthens, weakens, or even prunes synapses (*Kavanau, 1996*; *Datta et al., 2008*; *Chauvette et al., 2012*; *de Vivo et al., 2017*; *Li et al., 2017*). Further, there is disagreement on whether sleep primarily enables correlation-based plasticity mechanisms such as long-term potentiation (LTP), or homeostatic forms of plasticity that serve the function of regulating overall synaptic strength to stabilize neuron and circuit function (*Frank and Cantera, 2014*; *Aton et al., 2014*; *Hengen et al., 2016*). Finally, it is unclear whether sleep is merely permissive for some forms of synaptic plasticity (*Torrado Pacheco et al., 2021*), or whether being asleep is by itself sufficient to induce synaptic changes without a preceding salient learning event (*Tononi and Cirelli, 2014*). Here, we use a combination of real-time sleep classification, ex vivo measurements of postsynaptic strength in two neocortical areas and two different cell types, and in vivo optogenetic monitoring of evoked thalamocortical transmission in primary visual cortex (V1), to ask whether sleep and wake states constitutively drive widespread synaptic plasticity within neocortical circuits.

One influential hypothesis, the synaptic homeostasis hypothesis (SHY), has motivated much work on the role of sleep in brain plasticity (*Tononi and Cirelli, 2014*). SHY proposes that memories are formed during wake when animals actively sample their environment, primarily through the induction

of Hebbian LTP-like mechanisms that causes a net potentiation of synapses. This process would saturate synapses if left unopposed (*Miller and MacKay, 1994*; *Abbott and Nelson, 2000*; *Turrigiano and Nelson, 2004*), and sleep is proposed to be an offline state that allows neurons to renormalize synaptic weights by downscaling synaptic strengths (*Tononi and Cirelli, 2014*). This renormalization is postulated to be a global process that affects all or most synapses in many brain regions (*Tononi and Cirelli, 2014*). Critically, SHY predicts that excitatory synapses should on average be stronger after a period of wake, and weaker after a period of sleep. A feature of non-rapid eye movement (NREM) sleep is slow wave activity (SWA), in which the electroencephalogram (EEG) expresses large, low-frequency waves (<4.5 Hz, *Dijk, 2009*). It is well established that SWA is high during NREM immediately after prolonged wake and progressively decreases as sleep pressure diminishes (*Dijk, 2009*). SHY additionally proposes that the size of slow waves is correlated with cortical synaptic strengths, and that the oscillation in SWA with sleep and wake is driven by sleep-dependent synaptic downscaling that then diminishes SWA (*Tononi and Cirelli, 2014*; *Tononi, 2009*).

Studies supporting SHY have found that the expression of proteins associated with synaptic potentiation are higher, axon-spine interfaces (a corollary of synaptic strength) are larger, and evoked transcallosal cortical responses increase in slope after a period of wake (*Vyazovskiy et al., 2008*; *de Vivo et al., 2017*; *de Vivo et al., 2019*; *Diering et al., 2017*). These studies were conducted on animals ranging from early postnatal (*de Vivo et al., 2019*) to juvenile (*Liu et al., 2010*; *Spano et al., 2019*) to adult (*Vyazovskiy et al., 2008*). Importantly, these changes were observed without introducing any salient learning experiences for the animal, suggesting that simply being asleep or awake reduces or increases net synaptic weights, respectively. In contrast, other recent studies have found that synaptic transmission is potentiated by sleep (*Chauvette et al., 2012*) or is unaffected by sleep deprivation (*Matsumoto et al., 2020*). If excitatory synapses indeed oscillate in strength across sleep and wake states, then the firing rate of individual neurons would be expected to oscillate as well; while some studies have found such oscillations (but with variable effects between brain regions and neuronal populations; *Vyazovskiy et al., 2009*; *Miyawaki and Diba, 2016*; *Miyawaki et al., 2019*; *Watson et al., 2016*), others have observed stable firing across periods of sleep and wake (*Hengen et al., 2016*; *Torrado Pacheco et al., 2021*).

Together, these studies paint a complex picture of the possible roles of sleep and wake states in modulating synaptic plasticity. Some discrepancies between studies are likely due to methodological differences; for instance, not all studies clearly differentiate between effects of sleep and circadian cycle, carefully classify sleep states and prior sleep history, or directly measured synaptic strengths. Here, we set out to carefully test the hypothesis that neocortical synaptic weights weaken during sleep and strengthen during wake, using both ex vivo slice physiology and in vivo monitoring of synaptic strengths and evoked spiking. We used in vivo behavioral state classification in real time to track the accumulation of sleep and wake, which allowed us to detect natural sleep- or wake-dense epochs that occurred within a defined 5 hr circadian window (zeitgeber time [ZT] 3–8). We then immediately cut slices and measured miniature excitatory postsynaptic currents (mEPSCs) from pyramidal neurons in V1 or prefrontal cortex (PFC; specifically, prelimbic and infralimbic cortex). We found that mEPSC amplitudes were stable across sleep- and wake-dense epochs in both brain regions. To extend this finding to evoked transmission, we used in vivo optogenetics to monitor thalamocortical synaptic drive to visual cortex across natural sleep-wake epochs. Again, we found that thalamocortical synaptic drive and evoked spiking were stable across even prolonged periods of sleep and wake. Together, our data show that neocortical synaptic strengths are remarkably stable across naturally occurring periods of sleep and wake, indicating that sleep does not drive widespread constitutive weakening of excitatory neocortical synaptic strengths.

## Results

### Sleep/wake behavior of juvenile Long-Evans rats

We wished to compare several functional measures of neocortical synaptic strength after natural periods of prolonged waking or sleeping. Previous studies in support of sleep-dependent regulation of synaptic strength in rodents have spanned a wide age range, from early postnatal to juvenile to adult (*de Vivo et al., 2019*; *Spano et al., 2019*; *de Vivo et al., 2017*; *Liu et al., 2010*;

*Vyazovskiy et al., 2008*). Here and in subsequent experiments, we used juvenile rats (postnatal days 25–31) to allow for high-quality endpoint slice recordings, and to ensure that our experiments were performed within a highly plastic developmental window when Hebbian and homeostatic plasticity are known to be pronounced (*Lambo and Turrigiano, 2013*; *Hengen et al., 2016*; *Smith et al., 2009*; *Espinosa and Stryker, 2012*).

To this end we first analyzed behavioral data from animals in this age range, to understand their natural rhythms of sleep and wake across zeitgeber time (ZT), and determine when they are likely to experience prolonged periods of sleep or wake (*Figure 1A–D*; see *Figure 1—figure supplement 1* for breakdown of states). We monitored electromyograms (EMGs), EEG, and video, and used standard approaches to classify sleep states into wake and rapid eye movement (REM) or NREM slow wave sleep, as described previously (*Hengen et al., 2016*; *Torrado Pacheco et al., 2019*; *Torrado Pacheco et al., 2021*; see Materials and methods). In some experiments, we also differentiated active and quiet wake, as noted below. As expected, animals on average slept more during the light phase (ZT 0–12, ~60% asleep) than during the dark phase (ZT 12–24, ~40% asleep, *Figure 1B,D*; *Frank and Heller, 1997*; *Frank et al., 2017*); however, there was considerable variability from day to day and animal to animal in when they experienced periods enriched in sleep or wake (defined as a 4 hr period with >65% of time in that state, *Figure 1A*, compare animals 1 and 2). Another prominent feature of sleep is the modulation of slow wave amplitude by prior sleep history (*Dijk, 2009*; *Tononi and Cirelli, 2014*). As expected, the average power in the delta band (0.5–4 Hz), a proxy for slow wave amplitude (*Dijk, 2009*; *Vyazovskiy et al., 2008*), rose as net time spent awake accumulated in the dark (*Figure 1C*), and dropped during the day when animals typically spent more time sleeping (*Figure 1B,C*). These changes in delta power over ZT are similar to those seen in adult rodents (*Leemburg et al., 2010*). Thus, these animals exhibit homeostasis in slow wave amplitude across sleep and wake cycles, which SHY suggests is causally linked to synaptic downscaling (*Tononi and Cirelli, 2014*; *Tononi, 2009*).

## Real-time sleep classification

To study the impact of sleep and wake on synaptic efficacy, it is necessary to disentangle sleep from circadian effects. Since prolonged natural periods of wake and sleep tend to occur during distinct circadian periods, many studies have accomplished this by sacrificing animals at the same ZT (typically several hours after lights on) and comparing animals that have slept naturally to those that were continuously kept awake via sleep deprivation (*Liu et al., 2010*; *Diering et al., 2017*; *de Vivo et al., 2017*; *de Vivo et al., 2019*). However, the animal-to-animal and cycle-to-cycle variability in the timing of enriched sleep means that even when animals are sacrificed at the exact same ZT, there are likely to be considerable differences in their recent history of sleep and wake (e.g. compare animals 1 and 2 several hours after lights on, red arrows in *Figure 1A*). This prompted us to develop an approach to classify sleep and wake in real time, so we could reliably detect concentrated periods of sleep or wake within a defined circadian window for each animal, and then sacrifice them to probe synaptic strengths ex vivo.

To classify behavioral state in real time, we collected EEG, EMG from the nuchal (neck) muscles, and video as they freely behaved in an enriched environment. The recording chamber included a littermate separated by a thin, transparent plexiglass wall with nose-poke holes to allow for social interactions, toys for stimulation and play, and food and water available ad libitum. The data were acquired, analyzed, and plotted all within a custom program, which uses canonical markers to classify behavioral state into NREM, REM, or wake (*Figure 2A*). Briefly, the program computes a delta/beta and a theta/delta ratio from the corresponding frequency bands in the EEG (delta: 0.5–4 Hz; theta: 5–8 Hz; beta: ~20–35 Hz). Large deflections in the EMG are z-scored and animal pixel movement is extracted from the video recording. The classifier reads in these variables and applies a simple semi-automated decision tree, which uses three manually adjustable thresholds (delta, theta, and movement thresholds). Given what is known about the probability of sleep transitions, the classifier then adjusts these thresholds based on recent state history (additional details in Materials and methods). With this technique, we could determine the sleep/wake state of the animal with an accuracy of 93.3% when benchmarked against manual coding (*Figure 2D*). All classifications made in real time were manually verified post hoc. We could specify the length and required sleep or wake density for an epoch to pass threshold; for this set of experiments we chose threshold values that were within the range used by others to test SHY (*Liu et al., 2010*; *Diering et al., 2017*; *Vyazovskiy et al.,*

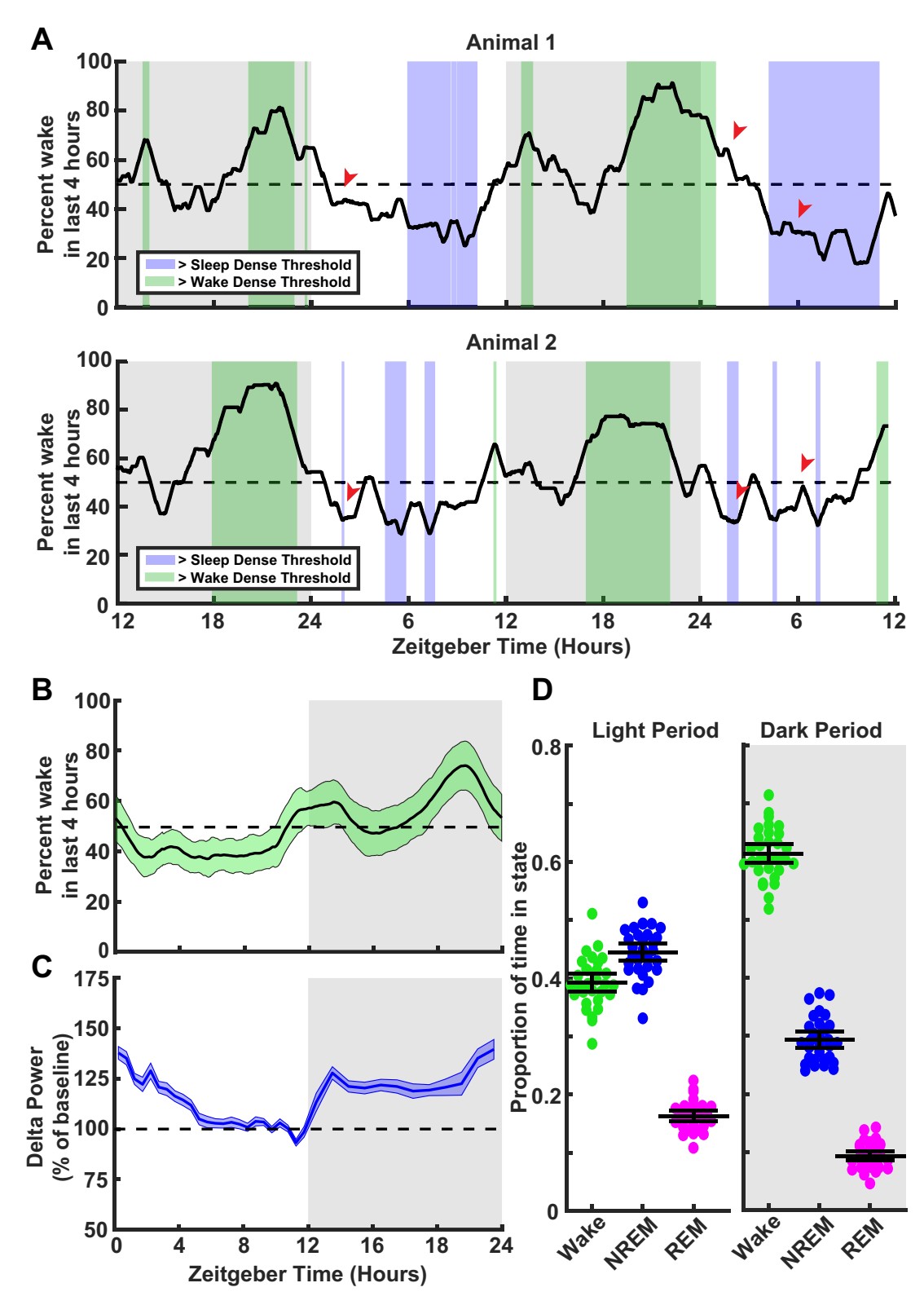

**Figure 1.** Characterization of sleep/wake behavior in juvenile Long-Evans rats. (**A**) Two example sleep/wake histories from two different animals. The y axis is a moving mean showing percent of time spent awake in the previous 4 hr; x axis is zeitgeber time (ZT) in hours. Periods where the animal is in a sleep-dense epoch (>65% time asleep in previous 4 hr) are colored blue, while >65% wake in previous 4 hr are colored green. Red arrows indicate points showing differing sleep/wake-dense experiences between animals at the same ZT. (**B**) Average percentage time spent awake in the previous 4 hr

*Figure 1 continued on next page*

*Figure 1 continued*

as a function of ZT. Light green shading indicates the standard deviation between animals; n = 30 animals. (C) Normalized delta power from the electroencephalogram (EEG) of animals in (B) as a percent of baseline (hours 7–11 in the light period). Shading represents SEM between animals. (D) Average time spent in each behavioral state shown as a proportion for each animal broken down by light period (left) and dark period (right); each point represents one animal. Mean and SEM shown, n = 30 animals.

The online version of this article includes the following figure supplement(s) for figure 1:

**Figure supplement 1.** Behavioral state breakdown over circadian cycle in juvenile Long-Evans rats.

**Figure supplement 2.** Stable sleep history in animals used in ex vivo slice experiments.

*2009*; *de Vivo et al., 2019*): 4 hr of >65% within a state, with the last hour >70% within the state. The 4 hr requirement was chosen because it is the longest span of time the rats naturally spent in a sleep-dense epoch in a typical day, and the last hour threshold was to ensure that synaptic differences were not rapidly reversed by recent behavioral state changes. The average density of sleep epochs that exceeded threshold and were included in the mEPSC analysis was 69%; the average wake density during the light period was 76% and during the dark period was 71%. NREM bouts were also more consolidated (longer) in the sleep-dense epoch as compared to wake (sleep 161.53 ± 10.4 s; light period, wake 120.97 ± 7.8 s; dark period, wake 122.16 ± 9.6 s; mean ± SEM).

We wished to compare three conditions: sleep-dense and wake-dense epochs that ended during a similar circadian period (ZT 3–8, *Figure 3A,B*), and wake-dense epochs that ended during the opposite circadian period (dark period, ZT 13–16, *Figure 3C*); this three-way comparison allowed us to look for differences driven by sleep/wake (sleep and wake at the same ZT) and differences driven by circadian time (wake at opposite ZTs). Spontaneous wake-dense epochs were infrequent in the light period, so when we detected a long natural wake epoch during the early light phase we added new toys to the chamber (e.g. red arrow in *Figure 3B*, top panel) to encourage additional waking. This was sufficient to extend natural waking epochs to reach threshold for wake dense within the specified circadian window (*Figure 3B*, *Figure 3—figure supplement 1*).

## Postsynaptic strengths are stable across extended periods of sleep or wake

In order to assess whether sleep induces a global downscaling of postsynaptic strengths, we tracked sleep/wake history in real time, cut acute slices once an animal finished a sleep- or wake-dense epoch (*Figure 3*), and performed whole-cell patch electrophysiology to record mEPSCs onto pyramidal neurons in two brain regions (V1 and PFC) and two layers (L4 and L2/3, see below for rationale for these targets). Ex vivo slice physiology is an established method to probe experience-dependent changes in synaptic strengths (*Khurana and Li, 2013*; *Lambo and Turrigiano, 2013*; *Nataraj et al., 2010*; *Heynen et al., 2003*; *Miska et al., 2018*), and mEPSC amplitudes are a standard readout of synaptic up- or downscaling and are a direct correlate of postsynaptic strength (*Turrigiano, 2008*). The sleep history (expressed as hypnograms that show the time in each sleep/wake state) of each animal included in our mEPSC analysis is shown in *Figure 3A–C*, top traces, and in *Figure 3—figure supplement 1*. Another way of visualizing an individual animal's sleep/wake history is to generate a moving average of the percent of time spent awake in the last 4 hr (*Figure 3A–C*, bottom traces; *Figure 3—figure supplement 1*); this illustrates when in ZT each animal passed the threshold for our endpoint mEPSC analysis. Using this approach we were able to measure mEPSC amplitudes at similar circadian time points from animals that had spent the majority of the proceeding 4 hr either awake or asleep, with minimal perturbation (*Figure 3A,B*); and from animals with similar wake experience at the opposite circadian time (*Figure 3C*).

We first recorded from L2/3 pyramidal neurons in V1 (*Figure 3D*). We chose visual cortex as it is highly plastic during this developmental window, which is within the classic visual system critical period (*Espinosa and Stryker, 2012*). Further, homeostatic mechanisms are present and are able to up- and downscale mEPSC amplitudes onto L2/3 pyramidal neurons during this developmental window (*Kaneko et al., 2008*; *Lambo and Turrigiano, 2013*; *Hengen et al., 2013*; *Torrado Pacheco et al., 2021*). When we computed the mean mEPSC amplitude by cell and compared this across conditions, we found no differences in amplitude after sleep-dense or wake-dense epochs in the light phase (sleep and L. wake), or wake-dense epochs in the dark period (D. wake; *Figure 3D,E*). Here

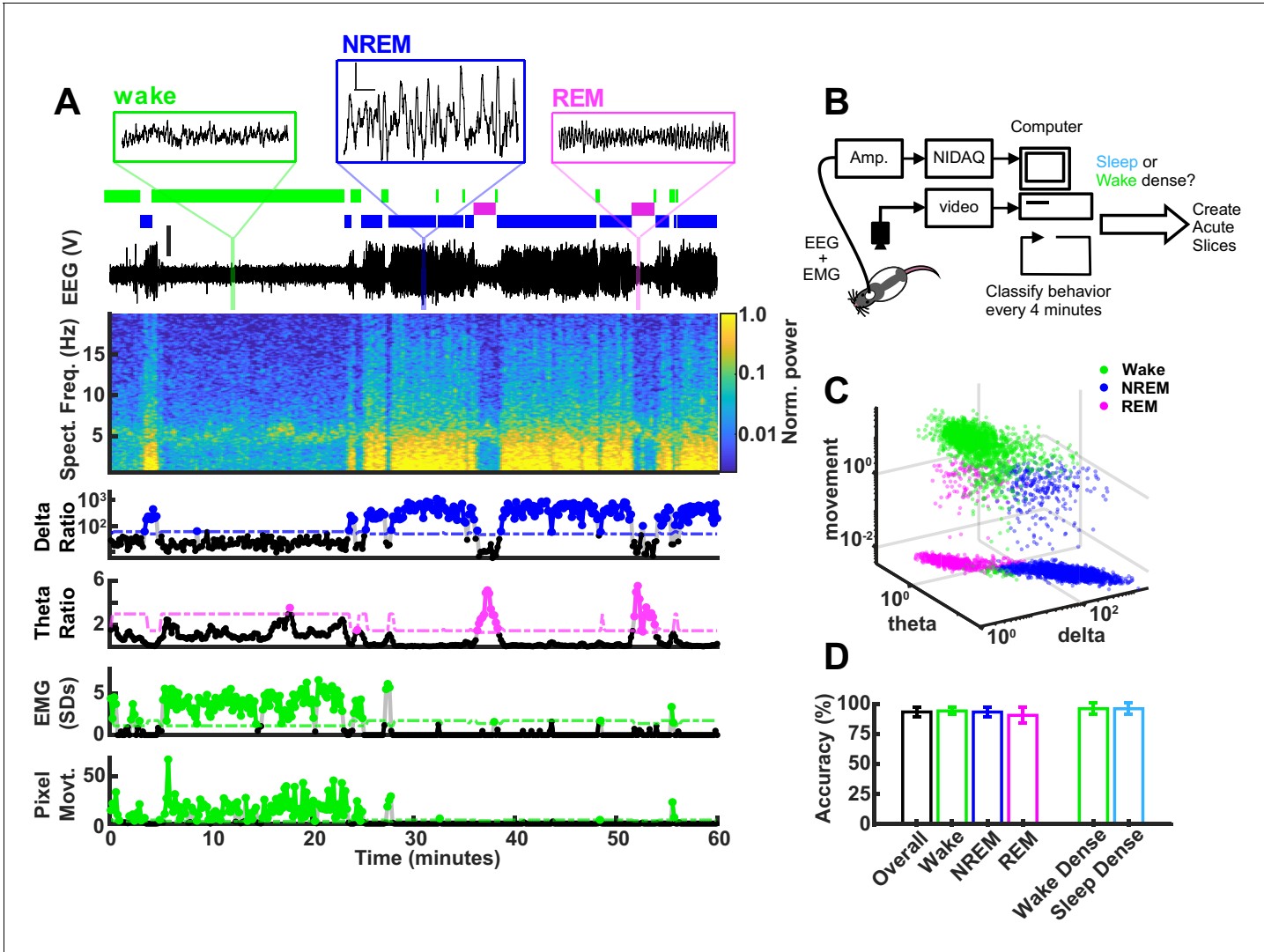

**Figure 2.** Real-time behavioral classification. (**A**) One hour of behavioral data illustrating variables used for classification of sleep/wake state, described in descending order. Top shows expanded raw electroencephalogram (EEG) traces from the indicated periods; scale bar next to non-rapid eye movement (NREM) example, 1 V, 0.5 s. Below expanded EEG is the sleep/wake classification expressed as a hypnogram, where colored bars indicate periods of wake (green), NREM (blue), or rapid eye movement (REM) (magenta). Beneath the hypnogram is the full raw EEG trace; scale is 2 V. Spectral frequency (Spect. Freq.) plots the full spectrogram of the EEG from 0 to 20 Hz, colored by power (normalized to maximum). Extracted features of the Spect. Freq. are plotted below: delta(0.5–4 Hz)/beta(20–35 Hz) power ratio (shown on log scale) is high during periods of NREM, while the theta(5–8 Hz)/delta(0.5–4 Hz) power ratio is high during periods of REM. Absolute electromyogram (EMG) values were normalized and expressed as standard deviation. Finally, animal movement in pixels is plotted. The bottom four plots of vigilance state features each have an adjustable threshold for state classification, shown as dashed line; points above the threshold are shown as colored dots. (**B**) Schematic of real-time classifier rig. (**C**) 3D plot of vigilance state features: delta ratio, theta ratio, and movement measures (zero movement assigned lowest observed value for log axis) colored by brain state. Data from example animal. (**D**) Accuracy of real-time classifier compared to manual scoring by state: overall (% time matching between all three states)=93.3%; wake = 94.2%; NREM = 93.2%; REM = 90.5%; wake dense (matching wake dense on and off times)=96.0%; sleep dense = 96.0%. The online version of this article includes the following figure supplement(s) for figure 2:

**Figure supplement 1.** Discrimination of active and quiet wake.

and below, cellular properties such as input resistance and resting potential were not different between conditions; mEPSC kinetics (rise times, decay tau, and scaled waveforms) were also not different between conditions. To examine more closely the mEPSC amplitude distribution in the different conditions, we randomly selected the same number of events from each neuron and plotted individual mEPSC amplitudes as a cumulative distribution (*Figure 3E*, right). There were no significant differences in the amplitude distribution between conditions. Thus neither prior sleep history

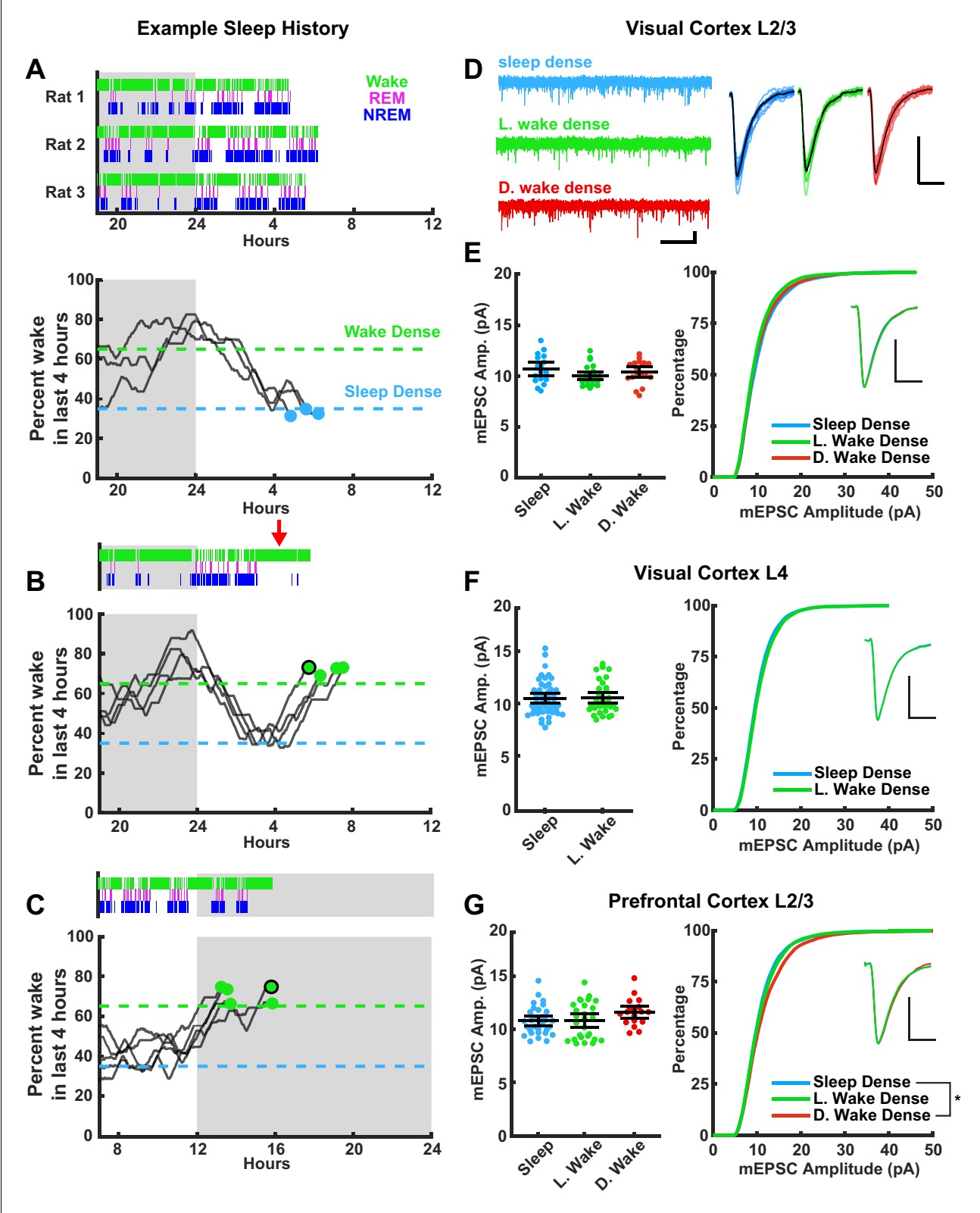

**Figure 3.** Stability of miniature excitatory postsynaptic current (mEPSC) amplitude after prolonged periods of sleep or wake. The left column represents example sleep histories for the VC L2/3 experiments while right column shows mEPSC characteristics for the indicated cell types and brain regions. (**A**) Top figure, hypnograms for three animals used for sleep-dense mEPSC recordings. Green bars indicate instances of wake, magenta indicates rapid eye movement (REM), and blue non-rapid eye movement (NREM). Bottom, this same behavioral state data represented as a moving average of % time

*Figure 3 continued on next page*

*Figure 3 continued*

spent awake in the preceding 4 hr; the dashed lines represent the thresholds for achieving the required density in a given state, and the dots represent endpoints for slice recordings. (B) and (C) are same as for (A), but for wake-dense recordings performed during the light and dark periods, respectively. Red arrow indicates wake encouragement (introduction of toy). Black circle around endpoint dot indicates which trace is represented by example hypnogram. (D) Traces on the left, example mEPSC recordings from L2/3 of primary visual cortex (V1); scale bar, 10 pA, 1 s. On the right, cell average mEPSC events; scale bar, 5 pA, 5 ms. (E) On left, cell average mEPSC amplitudes from L2/3 V1; error bars = 95% CI of mean (p=0.17; Kruskal-Wallis). (E) On right, cumulative histogram of sampled mEPSC amplitudes (SD – L. WD p=0.075, L. WD – D. WD p>0.9, SD – D. WD p=0.77; Kolmogorov-Smirnov (KS) test with Bonferroni correction). Inset for (E)–(G) is a peak scaled average mEPSC waveform; scale bar, 50% peak, 5 ms. Sleep dense n = 17 cells, three animals; L. wake dense n = 25 cells, four animals; D. wake dense n = 18 cells, five animals. (F) On left, cell average mEPSC from L4 V1 (p=0.52; Kruskal-Wallis). (F) On right, cumulative mEPSC amp. from L4 V1 (p=0.99; KS test). Sleep dense n = 47 cells, six animals; L. wake dense n = 32 cells, four animals. (G) On left, cell average mEPSC amplitudes from L2/3 prefrontal cortex (PFC) (p=0.13; Kruskal-Wallis). (G) On right, cumulative histogram of sampled mEPSC amplitudes (SD – L. WD p>0.9, L. WD – D. WD p=0.096, SD – D. WD p=0.015; KS test with Bonferroni correction). Sleep dense n = 31 cells, six animals; L. wake dense n = 28 cells, five animals; D. wake dense n = 18 cells, six animals.
The online version of this article includes the following figure supplement(s) for figure 3:

**Figure supplement 1.** Sleep/wake histories for additional ex vivo slice recordings.
**Figure supplement 2.** Drop in delta power and time spent awake do not correlate with miniature excitatory postsynaptic current (mEPSC) amplitudes or frequency.

nor circadian time significantly modulated mEPSC amplitude onto visual cortical L2/3 pyramidal neurons.

We next examined mEPSCs onto pyramidal neurons in L4 of V1. L4 pyramidal neurons receive extensive monosynaptic input from thalamic relay neurons, which vary their firing patterns dramatically during sleep (*Steriade, 2001*; *Steriade and Timofeev, 2003*), and these thalamocortical synapses are also highly plastic at this time (*Cooke and Bear, 2010*; *Miska et al., 2018*). Similar to our observations in L2/3, mEPSC amplitudes were indistinguishable after wake-dense or sleep-dense epochs, and there was no difference in the cumulative amplitude distribution between conditions (*Figure 3F*).

To determine whether this stability in postsynaptic strength was evident outside of primary sensory cortex, we next examined mEPSCs onto L2/3 pyramidal neurons in PFC. The PFC shows a strong modulation of SWA (*Leemburg et al., 2010*), has been suggested to be especially impacted by extended wake (*Muzur et al., 2002*; *Jones and Harrison, 2001*), and a previous study reported weaker synapses within PFC after sleep (*Liu et al., 2010*). However, consistent with our findings in V1, mean mEPSC amplitude was also stable across sleep- and wake-dense epochs in PFC and was not significantly different after extended wake at opposite circadian periods (*Figure 3G*, left). The cumulative amplitude distributions following extended sleep or wake in the light phase were indistinguishable (*Figure 3G*, right), but interestingly there was a small but significant rightward shift at the high end of this distribution in the D. wake condition (SD − D. wake, p=0.015). This suggests that there is a small circadian modulation of mEPSC amplitude onto L2/3 pyramidal neurons in PFC.

We next examined the frequency of mEPSC events. mEPSC frequency can be influenced by many factors (*Han and Stevens, 2009*; *Wierenga et al., 2006*) and is generally more variable between neurons than amplitude (*Lambo and Turrigiano, 2013*; *Hengen et al., 2013*; *Liu et al., 2010*; *Figure 4*). We found no significant differences in mean mEPSC frequency across any of the conditions or brain regions examined here (*Figure 4*). Inter-event intervals were not different between conditions in visual cortex L2/3 or L4. In PFC inter-event intervals were shorter in L. wake compared to both D. wake and L. sleep; thus there may be some interaction between time of day and sleep/wake state on mEPSC frequency in PFC, which does not generalize to visual cortex.

Because we have the full behavioral history of each animal prior to preparing brain slices, we could examine potential influences on mEPSCs beyond the duration of sleep or wake. There was a reasonable range of wake densities (from 65% to 90%) within the wake-dense condition, which allowed us to evaluate possible correlations between wake density and amplitude or frequency of mEPSCs; neither of these were significantly correlated (*Figure 3—figure supplement 2*). Further, we could test an important proposed relationship between changes in slow wave amplitude and excitatory synapses: according to SHY, a larger reduction in slow wave amplitude (i.e. decrease in delta power) should correlate with weaker synapses (*Tononi and Cirelli, 2014*; *Tononi, 2009*). However, when we plotted the decrease in delta power during prior sleep against the endpoint mEPSC

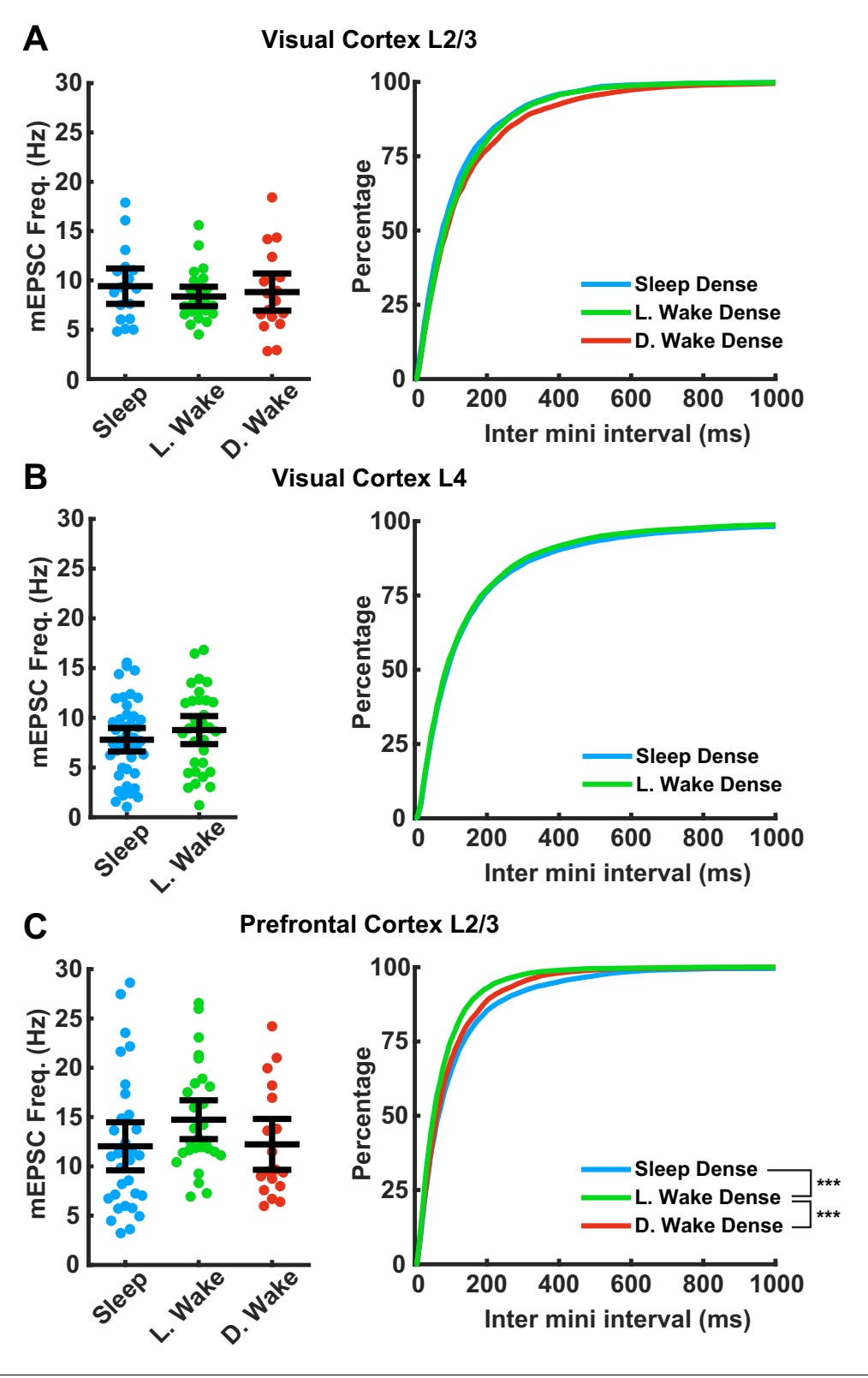

**Figure 4.** Impact of prolonged sleep/wake on miniature excitatory postsynaptic currents (mEPSC) frequency. mEPSC frequency is not consistently modulated by sleep/wake history across brain regions. (**A**) Data from visual cortex L2/3. Left plot, cell average mEPSC frequencies for the indicated conditions; error bars = 95% CI of mean (p=0.75, Kruskal-Wallis). Right plot, cumulative plot of inter-event interval (SD − L. WD p=0.35, SD − D. WD

*Figure 4 continued on next page*

*Figure 4 continued*

p=0.15, L. WD − D. WD p>0.9; Kolmogorov-Smirnov (KS) test with Bonferroni correction). Sleep dense n = 17 cells, three animals; L. wake dense n = 25 cells, four animals; D. wake dense n = 18 cells, five animals. (B) Same as in (A) but data from visual cortex L4; cell average on left (p>0.9, Kruskal-Wallis); cumulative plot on right (p=0.45, KS test). Sleep dense n = 47 cells, six animals; L. wake dense n = 32 cells, four animals. (C) Same as in (A) but data from prefrontal cortex L2/3. Left, cell average values for the indicated conditions (p=0.056, Kruskal-Wallis). Cumulative plot of inter-event interval (SD − L. WD p<1e-6, SD − D. WD p=0.74, L. WD − D. WD p=0.007; KS test with Bonferroni correction). Sleep dense n = 31 cells, six animals; L. wake dense n = 28 cells, five animals; D. wake dense n = 18 cells, six animals.

amplitude or frequency, we again found no relationship (*Figure 3—figure supplement 2A,B*). Together, our data show that postsynaptic strength in a range of excitatory neocortical cell types is remarkably stable across prolonged periods of sleep or wake.

## Thalamocortical evoked field EPSPs are stable over the course of sleep and wake

mEPSC amplitude is a useful measure of postsynaptic strength, but does not take into account potential changes in presynaptic function, and cannot be used to follow synaptic strengths prospectively across sleep and wake epochs. To complement the mEPSC measurements, we therefore designed a paradigm to measure functional synaptic strength continuously in vivo across sleep and wake epochs in freely behaving animals. In order to evoke synaptic responses in visual cortex from a defined population of inputs, we took advantage of our ability to express ChR2 in thalamocortical terminals within V1 (*Miska et al., 2018*) by virally expressing ChR2 (AAV-ChR2-mCherry) in dorsal lateral geniculate nucleus (dLGN) of the thalamus, which projects extensively to L4 of V1. After viral injection into dLGN, animals were given 1.5–2 weeks to allow for expression and transport of ChR2 to thalamocortical axon terminals in V1 (*Figure 5A*), and we then implanted linear 16-channel silicon probes into V1 with adhered optic fibers that rested on the cortical surface. With this configuration, we could reliably evoke thalamocortical responses with 1 ms pulses of blue light in freely behaving animals (*Figure 5B–E*). The linear design of the probe with exact electrode spacing, combined with current-source density analysis (*Figure 5B*) and post hoc histology, allowed us to estimate the placement of electrical sites with high confidence. These evoked responses were similar to those evoked by electrical stimulation of LGN, as well as visually evoked potentials (*Cooke and Bear, 2010*; *Niell and Stryker, 2008*). The amplitude of the first negative trough of this evoked response has the shortest latency in L4 (*Figure 5B–D*), where it arises primarily from monosynaptic thalamic excitatory synapses onto L4 neurons (*Khibnik et al., 2010*; *Cooke and Bear, 2010*), and thus represents an evoked field excitatory postsynaptic potential (fEPSP). We monitored and analyzed the amplitude and slope of this short latency fEPSP (*Figure 5B–G*, *Figure 6—figure supplement 1*) as a readout of functional thalamocortical synaptic efficacy in L4.

To chronically monitor the amplitude of thalamocortical fEPSPs, we used light intensities below the peak saturation of responses (~50% of the peak response), and very low stimulation frequencies (1/20–1/40 Hz) to avoid phototoxicity or the induction of plasticity from the stimulation itself. With these parameters, we were able to measure stable fEPSPs continuously for several days (*Figure 5E*), while simultaneously monitoring the sleep/wake state of the animal. An interesting feature of these recordings is that fEPSP amplitude was rapidly modulated by behavioral states (*Figure 5E,F*), as previously reported (*Vyazovskiy et al., 2008*; *Matsumoto et al., 2020*). To more fully capture behavioral state modulation of the fEPSP, we further classified wake into quiet wake and active wake using local field potential (LFP) and video data (*Figure 2—figure supplement 1*). We found that evoked fEPSPs were largest on average during NREM, of similar size during quiet wake and REM, and smallest during active wake (*Figure 5F*). These changes in fEPSP amplitude manifested rapidly upon transitions between states (*Figure 5E*), consistent with a rapid neuromodulatory effect (*Lee and Dan, 2012*; *Brown et al., 2012*; *Matsumoto et al., 2020*). A second feature of these fEPSPs is that they were of consistently higher amplitude in all behavioral states in the light than in the dark; there were no obvious circadian oscillations beyond a rapid change in amplitude at the transitions between light and dark periods (*Figure 5G*).

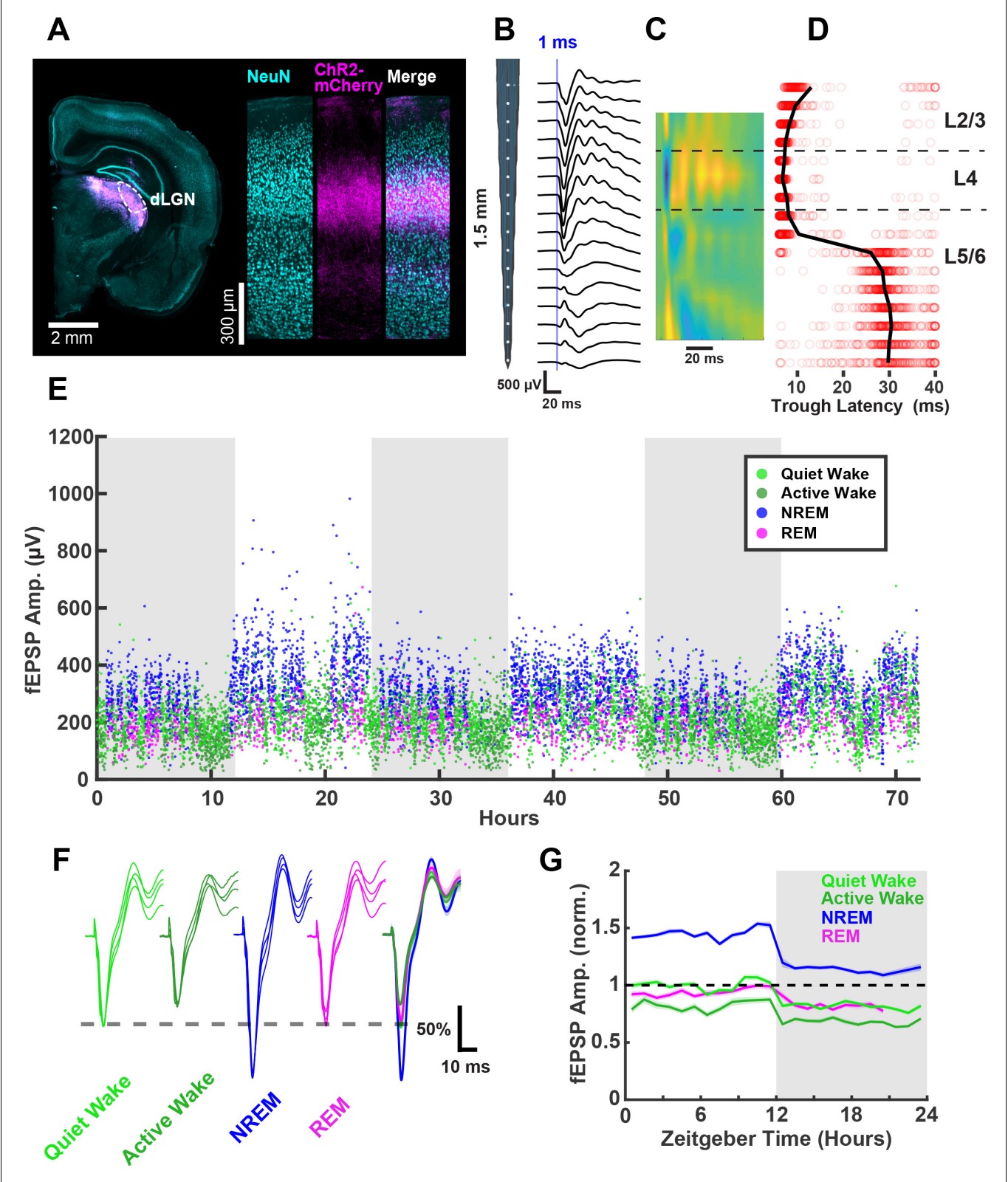

**Figure 5.** Thalamocortical (TC) evoked synaptic responses in freely behaving animals. (**A**) AAV-Chr2-mCherry viral expression targeted to dorsal lateral geniculate nucleus (dLGN) projects to V1. Left, coronal slice showing viral expression in magenta. Right, visual cortex showing neuronal nuclei marked with NeuN and ChR2-mCherry in magenta. (**B**) Evoked field excitatory postsynaptic potentials (fEPSPs) recorded with a linear silicon array show clear layer-specific waveforms consistently with TC afferent location. (**C**) Current source density heat plot identifies layers; cool colors indicate current sink. (**D**)

*Figure 5 continued on next page*

*Figure 5 continued*

fEPSP latency to first negative peak identifies layers. Red circles represent latency of individual events; black line represents mean for each channel. (E) fEPSPs recorded over 3 days in a single freely behaving animal. Amplitude of the response plotted and colored by brain state. Dark periods (lights off) are represented by gray rectangles. (F) fEPSP waveforms by brain state, where each waveform is from a different animal; normalized to QW peak (indicated by dashed line) for each animal. Scale bar, 50% of peak amplitude, 10 ms. (G) fEPSP amplitudes across circadian time, normalized to quiet wake in the light period. n = 4 animals.

SHY predicts that there should be a gradual decrease in the size of the fEPSPs during sleep, while wake should produce a gradual increase (*Vyazovskiy et al., 2008*). To test this, we evoked fEPSPs continuously at low frequency, identified sleep- or wake-dense epochs (using either a >65% or >75% time-in-state threshold, *Figure 6A*), and separately measured the fEPSP amplitude during each of the four behavioral states (REM, NREM, active wake, and quiet wake; *Figure 6*). We then plotted fEPSP amplitude across sleep-dense epochs measured either in NREM or REM (*Figure 6B,C*) and across wake-dense epochs measured in active or quiet wake (*Figure 6D,E*). Using a 65% threshold (*Figure 6B–E*, left column), we were able to isolate natural sleep-dense epochs of up to 4.75 hr and wake-dense epochs of up to 12.5 hr in duration; with a more stringent >75% threshold (*Figure 6B–E*, right column), these durations were somewhat shorter; up to 2.75 hr for sleep dense and 9 hr for wake dense.

Strikingly, we saw no progressive changes in fEPSP amplitude during prolonged sleep (*Figure 6B,C*) or wake (*Figure 6D,E*) using either density threshold, even for the longest duration epochs. This stability of the responses was evident regardless of which state we measured fEPSPs in *Figure 6*. As fEPSP amplitude estimates can be contaminated by polysynaptic or spiking activity occurring after the evoked monosynaptic thalamocortical synaptic potentials, we repeated this analysis using a second standard measure of fEPSP magnitude, the slope of the initial downward deflection. Consistent with the amplitude measurements, the slopes did not change across sleep and wake states (*Figure 6—figure supplement 1*).

As a second means of probing for sleep- or wake-induced changes in the fEPSP, we designed a 'sandwich' method for detecting changes across extended periods of sleep, by measuring the amplitude in flanking periods of wake separated by an extended period of sleep, and vice versa for extended periods of wake (*Figure 7A*); all measurements were made in NREM for flanking periods of sleep, and quiet wake for flanking periods of wake, and the results were identical when we instead used REM and active wake, respectively. For this analysis, we defined dense epochs as >30 min periods when the animal was >75% asleep/awake and identified all such epochs for each animal. Using this sandwich approach, we again found that fEPSP amplitudes were largely stable across periods of sleep or wake (*Figure 7B,C*; <2% change), indicating that the intervening periods of sleep and wake produce no persistent change in synaptic strength. When we plotted the change in fEPSP amplitude against the duration of the intervening sleep or wake epoch, we found no significant correlation between epoch duration and change in fEPSP amplitude (*Figure 7D,E*). Taken together with our mEPSC data, these experiments show that synaptic efficacy onto L4 excitatory neurons is stable across extended periods of sleep and wake.

## The ability of thalamocortical inputs to evoke spikes is stable over the course of sleep and wake

Although thalamocortical fEPSP amplitudes were not modulated by time spent awake or asleep, it is possible that the ability of thalamocortical synapses to evoke spiking in postsynaptic V1 neurons – a readout of functional connection strength – could be modified by prolonged periods of sleep or wake. In addition to recording fEPSPs, our optrodes allowed us to sample spikes in V1 evoked by stimulation of thalamocortical inputs (*Figure 8A*). Using spike sorting as described (*Hengen et al., 2016*; *Torrado Pacheco et al., 2021*), we were able to follow spiking in individual regular-spiking (putative pyramidal) neurons over the course of sleep and wake epochs. Although the probability of evoking a spike for any given stimulus was low, when averaged across many stimuli we found that most well-isolated units had a detectable response to the thalamocortical stimulation (*Figure 8A,B*). Consistent with the rapid brain-state modulation of fEPSP amplitude, there were more evoked spikes on average during NREM, an intermediate number during REM, and fewer during wake (*Figure 8B*). To determine whether evoked firing changed as a result of time spent asleep or awake,

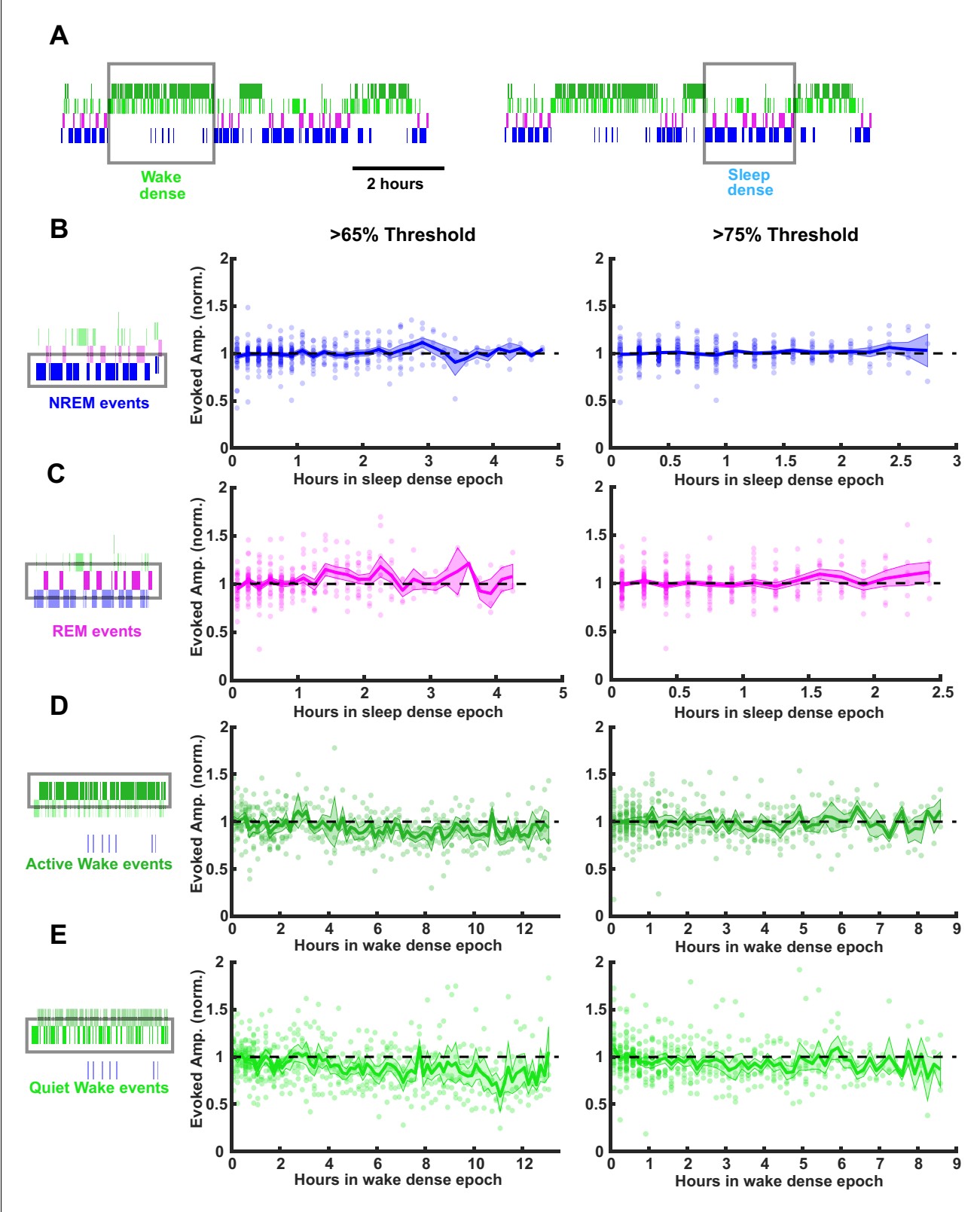

**Figure 6.** Stability of thalamocortical field excitatory postsynaptic potential (fEPSP) amplitude during prolonged periods of sleep or wake. Time series data of thalamocortical evoked fEPSPs over the course of either sleep- or wake-dense epochs. (**A**) Example hypnograms showing wake- (left) and sleep- (right) dense epochs. Gray rectangles indicate times in which stimulations were analyzed for this example. (**B**) fEPSP amplitudes during sleep-dense epochs, measured in non-rapid eye movement (NREM) (as illustrated by box around hypnogram on left). Epochs passing the >65% threshold are shown

*Figure 6 continued on next page*

*Figure 6 continued*

on left, and those passing the >75% threshold are shown on right. fEPSP amplitude was normalized to the beginning of each epoch. Each dot is data from one epoch, averaged over 10 min bins; n = 4 animals. Blue line = mean, shading = SEM, black dotted line = 1 (i.e. no change). (**C**) Same as (**B**) but for rapid eye movement (REM) events. (**D**) Same as (**B**) but normalized fEPSP amplitude during wake, measured during active wake. (**E**) Same as (**D**) but for quiet wake events.

The online version of this article includes the following figure supplement(s) for figure 6:

**Figure supplement 1.** Stability of field excitatory postsynaptic potential (fEPSP) slope during prolonged periods of sleep or wake.

we used the same 'sandwich' approach we used for the fEPSP data to compare the magnitude of evoked firing in flanking periods of wake separated by an extended period of sleep, and vice versa for extended periods of wake (*Figure 8C–E*). We averaged the response for each isolated unit across all sleep- or wake-dense epochs (defined as >30 min periods when the animal was >75% asleep/ awake), and found no significant differences in evoked spiking (*Figure 8E*).

Next, we took advantage of the variation in epoch durations to look for a correlation between the length of an epoch and the change in number of evoked spikes. We plotted the change in evoked spikes for each unit for each epoch, and found no significant correlation between duration of a sleep/wake epoch and ability of the thalamocortical stimulation to elicit spikes. Taken in concert with the fEPSP results, this shows that time spent asleep or awake does not cause progressive changes in the efficacy of thalamocortical synaptic transmission in vivo.

## Discussion

It has been postulated that patterns of brain activity during sleep induce a widespread downscaling of excitatory synaptic strengths throughout CNS circuits, but the evidence for this has been contradictory. Here, we used three complementary measures of functional synaptic strengths, coupled with real-time sleep classification in freely behaving animals, to ask whether simply being asleep is able to constitutively drive widespread synaptic weakening. We found that mEPSCs onto L4 or L2/3 pyramidal neurons were stable across sleep- and wake-dense epochs in both V1 and PFC. Further, chronic monitoring of thalamocortical synaptic efficacy in V1 of freely behaving animals revealed remarkable stability of thalamocortical transmission across prolonged natural sleep and wake epochs. Together, these data provide strong evidence that sleep does not drive widespread constitutive weakening of neocortical excitatory synaptic strengths.

Several previous studies have reported that molecular or morphological correlates of synaptic strength are lower after a period of sleep than a period of wake (*Vyazovskiy et al., 2008*; *de Vivo et al., 2017*; *Diering et al., 2017*). There are several methodological differences between our approach and previous studies that might explain some of these differences. First, we carefully classified the sleep history of every animal used for our ex vivo recordings, which allowed us to control for the considerable variability across individual animals in sleep habits. In the young animals studied here (4–5 weeks of age), the average chance of having experienced a sleep-dense epoch at ZT 4 or 6 (time points used in many studies) is <50%; our approach allowed us to wait until a sleep- or wake-dense epoch was detected within a specified circadian window for each animal to ensure consistency in sleep history. Second, we carefully controlled for circadian time, which has often been used as a proxy for sleep and wake, despite the potential impact of circadian time itself on synaptic strengths (discussed in *Frank and Cantera, 2014*; *Bridi et al., 2020*). Third, we used direct electrophysiological measurements of synaptic strength across largely unperturbed natural sleep and wake cycles, whereas many previous studies used indirect measures of synaptic function as proxies for synaptic strength (*Diering et al., 2017*; *de Vivo et al., 2017*), and/or relied on extended sleep deprivation paradigms to manipulate the amount of sleep (*Diering et al., 2017*; *de Vivo et al., 2017*; *Vyazovskiy et al., 2008*; *Liu et al., 2010*; *Khlghatyan et al., 2020*). Finally, there is wide variation in brain areas and cell types examined across studies. Correlates of synaptic weakening after sleep have been reported in motor cortex, somatosensory cortex, and PFC (*Liu et al., 2010*; *Vyazovskiy et al., 2008*; *de Vivo et al., 2017*; *Diering et al., 2017*), while here we examined synaptic strengths in visual cortex and PFC. It is possible that the impact of sleep on synaptic function is distinct in different brain areas and cell types (discussed in *Frank and Cantera, 2014*). Regardless,

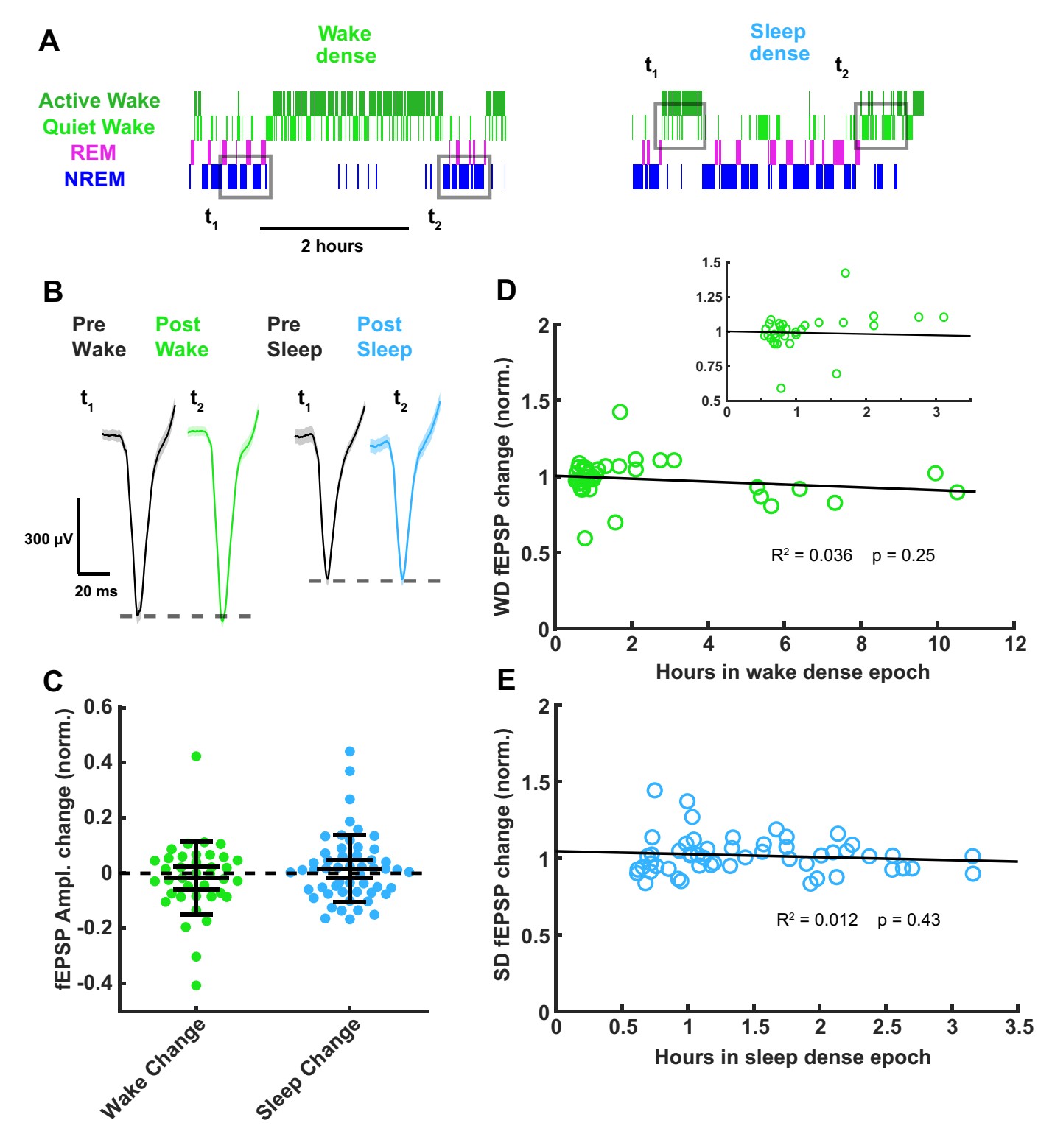

**Figure 7.** Thalamocortical field excitatory postsynaptic potential (fEPSP) amplitude measured before and after prolonged periods of sleep or wake. (A) Hypnograms showing an example of a wake- (left) and sleep- (right) dense epoch, with flanking periods of the opposite state. Gray rectangles indicate when measurements were taken for comparison. (B) Example fEPSP waveforms before and after a wake- (left) or sleep- (right) dense epoch. (C) Normalized change in fEPSP amplitude after wake or sleep (wake change, p>0.9; sleep change, p>0.9; Wilcoxon signed-rank test). (D) Correlative plot of duration of wake-dense epoch against normalized change in fEPSP amplitude. Black line = linear fit; $R^2$ value and p-value included on plot. Inset in

*Figure 7 continued on next page*

*Figure 7 continued*

top right is an expansion of plot between 0 and 3.5 hr. (**E**) Same as (**D**) but for sleep dense. n = 39 wake-dense epochs, 55 sleep-dense epochs, four animals.

our data show that synaptic downscaling is not a universal function of natural sleep across cortical cell types and brain regions.

Using a direct measure of postsynaptic strength, we found that mEPSC amplitude was stable across sleep and wake epochs in three cell types from two very different cortical areas, V1 and PFC. These findings are inconsistent with a previous study that reported that mEPSCs in frontal cortex (including the PFC region we recorded from in this study) were larger and more frequent after a period of sleep deprivation *Liu et al., 2010*; a third recent study in PFC found increased mEPSC amplitude but no change in frequency following sleep deprivation (*Khlghatyan et al., 2020*). While there are a number of possible explanations for these discrepancies between studies, one major difference is that here we more carefully controlled for sleep history in individual animals, and our study design allowed us to examine mEPSCs after relatively unmanipulated periods of natural sleep and wake that ended in the same circadian window. A second major difference is that we recorded from defined cell types and layers within well-defined brain regions, whereas previous studies combined recordings from several distinct regions and did not report cell type (*Liu et al., 2010*), or combined data from pyramidal neurons in several layers (*Khlghatyan et al., 2020*). As mEPSC frequency and amplitude can vary considerably between cell types, these measures could be strongly affected by sampling differences between conditions. By virtue of the EEG recordings performed in our experiments (generally absent in these previous studies for mEPSC datasets), we were also able to look for correlations in delta power drops and mEPSC amplitude, and were unable to see a significant relationship. An important question is whether mEPSC measurements are sensitive enough to detect changes in synaptic strength induced by 4 hr of consolidated sleep or wake. We have used a similar approach to successfully quantify changes in mEPSC amplitude induced by visual deprivation and eye reopening, and can reliably detect both increases and decreases of <10% with a similar sample size to the present study (*Lambo and Turrigiano, 2013*; *Hengen et al., 2016*; *Torrado Pacheco et al., 2021*). Previously reported changes in synaptic parameters following sleep or sleep deprivation are larger than that (*de Vivo et al., 2017*; *Liu et al., 2010*; *Vyazovskiy et al., 2008*), suggesting that if they were occurring at these synapses we would have been able to detect them. Taken together, our data show that natural periods of sleep and wake do not by themselves globally modulate postsynaptic strengths.

Whereas mEPSC amplitude strongly correlates with synaptic glutamate receptor accumulation and postsynaptic strength (e.g. *Ibata et al., 2008*; *Wierenga et al., 2005*), mEPSC frequency is influenced by many parameters and does not have a straightforward correlate with presynaptic function. The frequency of mEPSCs can be impacted by changes in the number of functional synaptic sites, but while some studies have found learning-dependent changes in spine turnover during sleep (*Li et al., 2017*), sleep does not constitutively drive changes in neocortical spine density (*de Vivo et al., 2017*). mEPSC frequency can also be modulated by factors that affect presynaptic excitability, including presence of neuromodulators (*Choy et al., 2018*; *Sharma and Vijayaraghavan, 2003*), and by vesicle refilling and pool size (*Zhou et al., 2000*; *Liu and Tsien, 1995*). Previous studies from PFC found inconsistent effects of sleep on mEPSC frequency; one study found increased mEPSC frequency after sleep deprivation (*Liu et al., 2010*) while another found no effect (*Khlghatyan et al., 2020*). While we found that mean mEPSC frequency was unaffected by sleep or wake in any condition, there were small shifts in inter-event interval distributions in some conditions. It is not clear if these effects are tied to circadian or sleep/wake differences, and they were not consistent across brain regions.

To complement our ex vivo mEPSC recordings, we also devised an approach that let us follow synaptic strength at a defined set of synaptic inputs in real time during naturally occurring periods of extended sleep or wake. We evoked thalamocortical fEPSPs at low frequency in L4 of V1 while monitoring vigilance state, and found that – like mEPSC amplitude – these evoked synaptic events were remarkably stable even across very long periods of sleep and wake. As for changes in mEPSCs, previous studies have reported a wide range of effects of sleep on evoked synaptic transmission. Sleep and/or SWA has been variously found to potentiate evoked responses (*Chauvette et al., 2012*) or

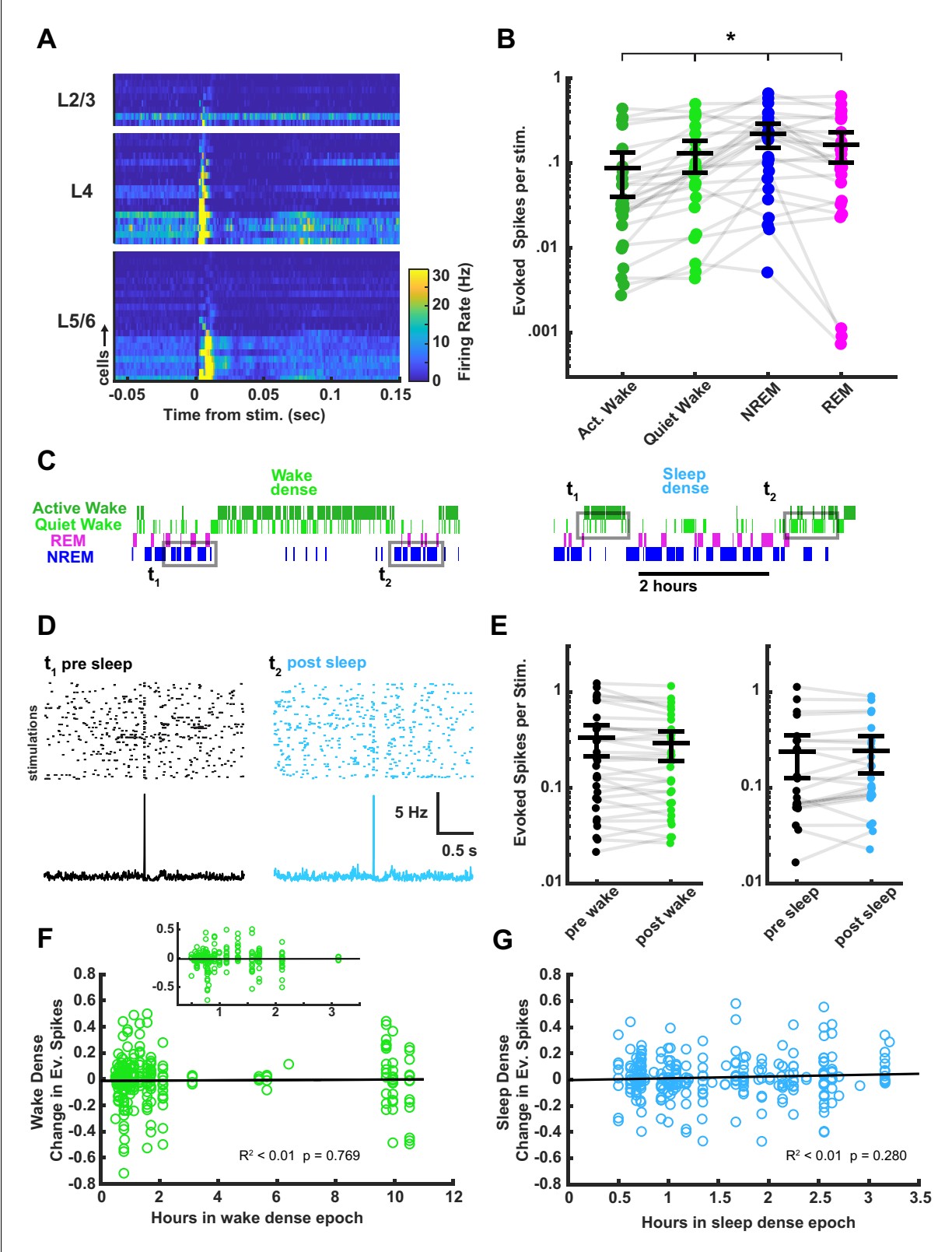

**Figure 8.** Characterization of thalamocortical evoked spiking in visual cortex between behavioral states and its stability across sleep and wake. (A) Heat plot response of the firing rate of each cell isolated during stimulation. Data are grouped by layer and sorted by the magnitude of response within each layer. n = 45 cells, four animals. (B) Evoked spikes per stimulation for each of the four vigilance states. The y axis represents the average number of evoked spikes for a single stimulation, calculated as spikes in a 15 ms post-stimulus window, minus the expected number of baseline spikes in that

*Figure 8 continued on next page*

*Figure 8 continued*

window; each dot represents a cell. Evoked spikes varied significantly by vigilance state (AW-QW, p=3.4e-4; AW-NREM, p=6.56e-4; AW-REM, p=4.70e-4; QW-NREM, p=0.0022; QW-REM, p=0.0122; NREM-REM, p=0.0380). (C) To determine whether evoked spike rates were affected by time spent awake or asleep, we compared evoked spiking before and after a sleep- or wake-dense epoch, using a >75% time in state threshold, as for field excitatory postsynaptic potentials (fEPSPs) in *Figure 7*. Hypnograms show examples of wake- or sleep-dense epochs with flanking periods of the opposite state. Gray rectangles indicate the stimulation times chosen for the analysis. (D) Example raster plots and post-stimulus time histograms of firing rate for an example cell pre-post sleep. (E) The number of evoked spikes before and after wake- (left) or sleep- (right) dense epochs; each dot is one cell, averaged across all of the wake- or sleep-dense epochs (wake dense n = 32 cells, four animals; sleep dense n = 24 cells, four animals). (F) Correlative plot of duration of wake-dense epoch against change in evoked spikes; each dot represents the response of a single cell to a single epoch (post – pre). Black line = linear fit; $R^2$ value and p-value included on plot. Inset is an expansion of plot between 0.25 and 3.5 hr. (G) Same as (F) but for sleep-dense epochs. n = 35 wake-dense epochs, 50 sleep-dense epochs, four animals.

bias evoked responses toward depression (*González-Rueda et al., 2018*; *Vyazovskiy et al., 2008*), while a recent study using optogenetically evoked responses in mouse motor cortex found no impact of sleep deprivation on evoked transmission (*Matsumoto et al., 2020*). Our approach (similar to *Matsumoto et al., 2020*) has the advantage of sampling thalamocortical synaptic strengths continuously over many iterations of nature sleep and wake cycles, and revealed no impact of time spent asleep or awake on evoked synaptic strengths in V1. One concern with this technique is that repeated optogenetic measurements from the same synapses might itself affect the synaptic responses, as high levels of blue light can affect neuronal health and RNA expression (*Tyssowski and Gray, 2019*). However, we used low light intensities and very low stimulation frequencies (1/20–1/40 Hz), so that total exposure to blue light was ~8000-fold less per hour than levels shown to impact function (*Tyssowski and Gray, 2019*); further, we were able to evoke consistent and stable responses over several days, suggesting that our stimulation paradigm was not inducing significant phototoxicity or otherwise compromising evoked transmission.

While time spent awake or asleep had no impact on evoked thalamocortical fEPSP responses, these responses were rapidly modulated by transitions between brain states, as has been noted previously (*Matsumoto et al., 2020*; *Vyazovskiy et al., 2008*; *Reinhold et al., 2015*; *Hall and Borbely, 1970*). This is likely due to rapid changes in neuromodulatory tone (*Lee and Dan, 2012*), and/or changes in the network activity in the thalamus or V1 (*Steriade, 2001*; *Steriade and Timofeev, 2003*) between sleep and wake states. The changes we observed in fEPSP amplitude correlated with changes in the ability of thalamocortical activation to evoke spikes in V1 neurons. Interestingly, thalamocortical transmission was most effective at evoking spikes during NREM sleep, with implications for information transfer during sleep states. We also found rapid modulation of thalamocortical efficacy at transitions between the light and dark phase of the 12/12 L-D cycle; as there were no other obvious circadian oscillations in efficacy, this is likely a rapid effect of changes in visual input. These data make clear that we can readily detect changes in thalamocortical synaptic efficacy driven by neuromodulatory or sensory input, indicating that the stability we observe during prolonged periods of sleep and wake is not due to a lack of sensitivity of our measurements.

Another method researchers have used to probe for sleep/wake-driven changes in excitability is to measure spontaneous firing of neurons in vivo across sleep and wake states. While some chronic long-term recordings have revealed an oscillation in frontal cortex firing rates consistent with changes in net excitatory drive (*Vyazovskiy et al., 2009*), other studies have either found variable effects between regions and across neuronal populations (*Miyawaki and Diba, 2016*; *Miyawaki et al., 2019*; *Watson et al., 2016*) or have observed stable firing rates absent a plasticity induction paradigm (*Hengen et al., 2016*; *Torrado Pacheco et al., 2021*), suggesting that the impact of sleep and wake on spontaneous firing is complex and brain area dependent. Spontaneous firing arises from many sources, so here we instead measured the ability of thalamocortical inputs to V1 to evoke spikes. Consistent with our findings that mEPSC amplitudes and thalamocortical fEPSPs are stable across extended periods of sleep and wake, we found that the functional connection strength between thalamus and V1 was also stable (*Figure 8*). Thus, in this study we used three complementary approaches to measure synaptic efficacy, and all three measures paint the same picture: that synaptic efficacy in V1 (as well as mEPSC amplitude in PFC) is not constitutively modulated by time spent awake or asleep. We note that this does not rule out a role for sleep in increasing or decreasing specific connections within specific brain circuits during specific developmental windows

(e.g. *Chauvette et al., 2012*; *Vyazovskiy et al., 2008*), but taken together our data show that synaptic downscaling is not a universal constitutive function of sleep.

While sleep is not sufficient to induce widespread changes in cortical synaptic strengths, there is a large body of work showing that sleep and wake states can profoundly affect the induction of various forms of plasticity when they are initiated by robust learning or sensory deprivation paradigms. For example, sleep facilitates and is required for several forms of visual system plasticity, and roles for NREM and REM sleep have been found in promoting growth, maintenance, and loss of specific synaptic connections during learning (*Frank et al., 2001*; *Aton et al., 2014*; *Yang et al., 2014*; *Li et al., 2017*; *Durkin et al., 2017*). Further, when firing in V1 is perturbed using visual deprivation/ eye reopening paradigms to induce homeostatic compensation, upward firing rate homeostasis is confined to periods of active wake (*Hengen et al., 2016*), while downward firing rate homeostasis is confined to periods of sleep (*Torrado Pacheco et al., 2021*). Upward and downward firing rate homeostasis are driven in part through homeostatic changes in synaptic strengths (*Hengen et al., 2013*; *Torrado Pacheco et al., 2021*), indicating that while natural wake and sleep episodes do not drive constitutive changes in V1 excitatory synaptic strengths, they are able to gate the induction of upward and downward homeostatic synaptic plasticity, respectively. Combined with this larger literature, our findings are most consistent with the view that sleep does not have a single unified effect on synaptic strengths, but rather that sleep and wake states are able to gate various forms of synaptic plasticity when they are induced by salient experiences.

# Materials and methods

## Key resources table

| Reagent type (species) or resource | Designation | Source or reference | Identifiers | Additional information |
| --- | --- | --- | --- | --- |
| Strain, strain background (*Rattus norvegicus*) | Long-Evans Rat | Charles River Labs | Charles River 006; RRID:RGD_2308852 | |
| Recombinant DNA reagent | AAV-ChR2(H134R) -mCherry | UPenn Vector Core | Penn ID: AV-9–20938M; Addgene: 100054-AAV9 | |

## Overview

All experimental measurements were performed on Long-Evans rats of both sexes between postnatal day P25 and P31. Litters were housed on a 12/12 light cycle with the dam and with free access to food and water. All animals were housed, cared for, surgerized, and sacrificed in accordance with Brandeis IBC and IACAUC protocols. Rat pups were weaned at ~P21. All animals received 2 days of post-operative care including daily injection of meloxicam and penicillin. During the few days before recording, animals were handled twice daily by the experimenter. The day before recording started, animals were transferred to a clear plexiglass cage of dimensions 12"×12"×16' (length, depth, height) and separated into two arenas by a clear plastic divider with 1' holes, which safely allows for tactile and olfactory interaction with a littermate that was present for all recordings. The recording chamber contained bedding, toys, and had walls with black/white edged patterns for enrichment. Animals were kept on a 12/12 L-D cycle in a temperature controlled room (21°C, 25–55% humidity). Animals in the dark period wake-dense condition (D. wake) were on an inverted L-D cycle, in order to keep all other experimental variables (time of slicing and recording) the same for the experimenter. To do this, the whole littler with dam were transferred to an L-D chamber with inverted light schedule (lights off 7:30 am to 7:30 pm) at least 2 weeks prior to surgery, and maintained on an inverted schedule while recording EEG/EMG.

## Virus injections into dLGN

Viral injections of AAV-CAG-ChR2-mCherry were performed between P12 and P16 using stereotaxic surgery under isoflurane anesthesia at 1.5–2.5%. dLGN was targeted bilaterally using stereotaxic coordinates after adjusting for the lambda-bregma distance for age. A glass micropipette pulled to a fine point delivered ~400 nL of virus-containing solution at the targeted depth. Targeting and expression was verified via post hoc histology.

## Electrode implantation surgeries

Animals underwent electrode implantation at ages P22–P26. For animals to undergo acute slice physiology, we implanted three EEG screws (PlasticsOne, Roanoke, VA): frontal lobe (B: +1–2, L: 1.5–2.5 mm), parietal lobe (B: −3–4, L: 1.5–2.5 mm), and cerebellum (for reference). In addition, two stainless steel spring electrodes were placed in the nuchal (neck) muscles of the rats for EMG recording. Screw electrodes were fed into a pedestal (PlasticsOne) and everything was secured with dental cement. For in vivo recording and stimulation experiments, a 16-channel silicon probe (Neuronexus, Ann Arbor, MI) with adhered optic fiber (200–400 μm diameter; Doric Lenses, Quebec City, Quebec; ThorLabs, Newton, NJ) were stereotaxically inserted into V1 in dLGN virus-injected animals such that the fiber tip rested on the surface of the brain. In the majority of these surgeries, light stimulation and recording were performed during the operation to verify correct placement of the optrode. The exposed craniotomy was surrounded with a silicone elastomer (Kwik-Cast, World Precision Instruments, Sarasota, FL). The array was secured using dental cement, then grounded to two screws (cerebellum and frontal lobe) using steel wire and soldering paste. Screws were not used for EEG data collection in the optrode animals as LFP was sufficient for behavioral classification. Animals were given 2–3 days of recovery before habituation to the recording chamber and data collection, at which point sleep/wake behavior had stabilized and was comparable to unsurgerized animals (see *Figure 1—figure supplement 2*).

## Real-time behavioral recording and classification

Animals with EEG/EMG head caps were connected to a flexible cable and low resistance commutator (PlasticsOne). EEG signal was recorded as the difference between the parietal and frontal screw, grounded to the cerebellum screw. The signal was amplified and filtered with EEG and EMG amplifiers at 5000× and 500×, respectively (BioPac, Goleta, CA). This analog signal was then digitized with a NIDAQ board (National Instruments, Austin, TX). The data were acquired, analyzed, and plotted all within a custom MATLAB program, which uses canonical markers to classify behavioral state as either NREM, REM, or wake. Briefly, the program computes a delta/beta and a theta/delta ratio from the corresponding frequency bands in the EEG (delta: 0.5–4 Hz; theta: 5–8 Hz; beta: ~20–35 Hz). Ratios were chosen as features because they normalize the power band measurements between animals. Because delta is high during NREM, and beta typically low (*Uchida et al., 1992*), taking the delta/beta ratio was logical and proved to be a stable and informative feature for classifying NREM. Likewise, theta is high during REM, and delta low, and therefore was indicative of REM sleep (ratios used in *Rempe et al., 2015*). EMG data is normalized to standard deviation and large deflections that cross a threshold (3–4 standard deviations) were extracted and considered as significant muscle activity. Animal pixel movement is extracted from the video recording (infrared light allowed video monitoring in darkness) using open-source software written in C++ (*Collins and Hashemi, 2019*; Video Blob Tracking, Open Source Instruments, Watertown, MA; Github: https://github.com/OSI-INC/VBT) and modified in-house to suit our needs (adding video cropping, tracking in both light and dark conditions). The program uses three manually adjustable thresholds (delta, theta, and movement thresholds) for classification, which are additionally adjusted given recent behavioral states. This moving threshold was chosen because the transition probabilities between different states are far from uniform (e.g. 10 s of NREM is more likely to follow prior NREM than wake; *Perez-Atencio et al., 2018*). Using known transition probabilities makes a scalar threshold more accurate. The classifier uses the EEG ratios and movement measures and applies a decision tree which predicts states in 10 s epochs given the thresholds. The recent behavioral data is plotted and added to a graph that contains the full behavioral history of the animal. This cycle repeats every 4 min. Animals were randomly assigned to either the sleep- or wake-dense conditions.

To extend wakefulness in light period wake-dense recordings, we generally waited until animals had experienced ~50% wake in the previous 4 hr, and then encouraged further wakefulness by moving, removing, or adding new toys or stirring the bedding; this procedure was generally initiated 1.25–2.25 hr before slicing and maintained until animals had reached criterion for wake density. It is important to note that all animals experience the removal and addition of new toys and changes to bedding regularly, so although the frequency of these manipulations is higher during this wake extension they are familiar procedures to the animals.

## Post hoc semi-automated behavioral state scoring

In order to perform the final classification of behavior after experiments, an in-house MATLAB/Python GUI was used. The GUI initially uses the threshold-based algorithm used in the real-time classifier for the first several hours, which is manually reviewed for accuracy. These data from each animal are then used as training input for the machine learning algorithm, a 200–300 tree random forest machine learning algorithm that uses the features described above and additional features found to be useful for the decision tree. These include the proportional contribution of delta, theta, and gamma powers; min/max EEG; and a past value of many of these features up to three 10 s bins ago. The random forest classifier was then applied and manually reviewed for the rest of the recording. While the automated classification step occurred in 10 s blocks, the subsequent manual review had a resolution of 1 s, ensuring our ability to capture particularly short microarousals. Random forest accuracy was roughly 97% benchmarked against manual coding.

In order to classify active vs. quiet wake, the same procedure was used except the training data included a manual separation of these states (*Figure 2—figure supplement 1*). Manual differentiation of active vs. quiet wake relied primarily on the video data (extracted movement measures and manual review of video segments to observe animal movement by eye), but also included use of gamma power. Animals were considered to be in active wake when LFP signals were especially desynchronized (high gamma power, low delta) and were physically moving around the cage/exploring. In contrast, quiet wake periods contained much less full-body displacement and were mostly composed of sitting or grooming/ingesting.

## NREM delta analysis

Delta power (0.5 Hz < Freq. < 4 Hz) was extracted from the EEG data of animals used for slice experiments, using FFT with 10 s windows. To calculate the change in delta power across ZT, we corrected for time-of-day differences in the amount of NREM (according to the approach of *Franken et al., 2001*). As delta power varies greatly between animals/recordings, we normalized it to the value at the end of the light period where it tended to be lowest (ZT 7–11). To calculate the change in delta power for each animal (*Figure 3—figure supplement 2*), we found the percentage drop from sleep epochs that occurred in the window ZT 23.5–1.5 to the last hour of sleep right before sacrificing animals.

## Ex vivo acute brain slice preparation

For brain slice preparation, animals between P25 and P31 were briefly anesthetized with isoflurane (usually under 60 s), and coronal brain slices (300 µm) containing V1 or PFC were obtained from both hemispheres of each animal. After slicing in carbogenated (95% $O_2$, 5% $CO_2$) standard ACSF (in mM: 126 NaCl, 25 $NaHCO_3$, 3 KCl, 2 $CaCl_2$, 2 $MgSO_4$, 1 $NaH_2PO_4$, 0.5 Na-ascorbate, osmolarity adjusted to 315 mOsm with dextrose, pH 7.35), slices were immediately transferred to a warm (34°C) chamber filled with a continuously carbogenated 'protective recovery' (*Ting et al., 2014*) choline-based solution (in mM: 110 choline-Cl, 25 $NaHCO_3$, 11.6 Na-ascorbate, 7 $MgCl_2$, 3.1 Na-pyruvate, 2.5 KCl, 1.25 $NaH_2PO_4$, and 0.5 $CaCl_2$, osmolarity 315 mOsm, pH 7.35) for 10 min, then transferred back to warm (34°C) carbogenated standard ACSF and incubated another 20–30 min. Slices were used for electrophysiology between 1 and 6 hr post-slicing.

## Whole-cell recordings

V1 and PFC were identified in acute slices using the shape and morphology of the white matter as a reference. Pyramidal neurons were visually targeted and identified by the presence of an apical dendrite and teardrop-shaped soma, and morphology was confirmed by post hoc reconstruction of biocytin fills. Borosilicate glass recording pipettes were pulled using a Sutter P97 micropipette puller, with acceptable tip resistances ranging from 3 to 5 MΩ. Cs + methanesulfonate-based internal recording solution was modified from *Xue et al., 2014*, and contained (in mM) 115 Cs-methanesulfonate, 10 HEPES, 10 BAPTA•4Cs, 5.37 biocytin, 2 QX-314 Cl, 1.5 $MgCl_2$, 1 EGTA, 10 $Na_2$-phosphocreatine, 4 ATP-Mg, and 0.3 GTP-Na, with sucrose added to bring osmolarity to 295 mOsm, and CsOH added to bring pH to 7.35. Inclusion criteria included $V_m$, $R_{in}$, and $R_s$ cutoffs as appropriate for experiment type and internal solution; values are listed below.

All recordings were performed on submerged slices, continuously perfused with carbogenated 34°C recording solution. Neurons were visualized on an Olympus upright epifluorescence microscope using a 10× air (0.13 numerical aperture) and 40× water -immersion objective (0.8 numerical aperture) with infrared differential interference contrast optics and an infrared CCD camera. Data were low-pass filtered at 5 kHz and acquired at 10 kHz with Axopatch 200B amplifiers and CV-7B headstages (Molecular Devices, Sunnyvale, CA). Data were acquired using an in-house program written either in Igor Pro (Wavemetrics, Lake Oswego, OR) or MATLAB (Mathworks, Natick, MA), and all post hoc data analysis was performed using in-house scripts written in MATLAB.

## mEPSC recordings

For spontaneous mEPSC recordings, pyramidal neurons were voltage clamped to −70 mV in standard ACSF containing a drug cocktail of TTX (0.2 μM), APV (50 μM), picrotoxin (25 μM); 10 s traces were obtained and amplified (10–20×). Event inclusion criteria included amplitudes > 5 pA and rise times < 3 ms. Neurons were excluded from analysis if $R_s$ >25 MΩ or $V_m$ > −50 mV.

## mEPSC analysis

To reliably detect mEPSC events above noise and limit bias in selection, we used an in-house program written in MATLAB that employs a semi-automated template-based detection method contained in a GUI (*Torrado Pacheco et al., 2021*). In brief, the program first filters the raw current traces and then applies a canonical mEPSC event- shaped template to detect regions of best fit. Multiple tunable parameters for template threshold and event kinetics that aid in detection were optimized and then chosen to stay constant for all analyses. Briefly, sample traces were chosen, which had all of the mEPSC events manually annotated. This training dataset was used while we sampled much of the parameter space for template-matching parameters. These parameters include the mEPSC shape used for template, how the error around the template was calculated, the threshold cutoff, and a coefficient that balances detection of larger mini events vs. smaller ones to correct for larger error in larger events. After identification, putative events were then examined for hallmark features of mEPSC (rise times, decay kinetics, conforming to amplitude cutoff, not saturated) and some detected events (~4%) were excluded if they did not conform to these features. Finally, the automatic detection missed ~6% of events and these were manually added. Results did not differ substantially when the data were not manually corrected.

## Array recording and stimulation

The arrays were connected to an Intan RHD2216 amplifier board, which in turn was connected to a RHD2000 SPI interface cable (Intan, Los Angeles, CA). The cable connected to a custom-designed Hall effect-based active commutator (NeuroTek Innovative Technology Incorporated, Toronto, Ontario, Canada). The commutator fed this digitized data into the RHD2000 USB board (Intan). Data were recorded at 25 kHz continuously for up to 100 hr using the RHD2000 Interface Software (Intan). Spike extraction, clustering, and sorting were done using custom MATLAB and Python code (see below). Light for stimulation was produced using a fiber coupled 470 nm LED system (M470F3/LEDD1B, Thorlabs). Light irradiance was adjusted for each animal and never exceeded 18 mW/cm². Stimulation was triggered using voltage pulses of varying magnitude from a NIDAQ board (controlled by custom MATLAB program) to the LED driver, set to modulation mode.

## Semi-automated spike extraction, clustering, and sorting

Spike extraction, clustering, and sorting was performed as previously described (*Hengen et al., 2016*; *Torrado Pacheco et al., 2021*). Spikes were detected in the raw signal as threshold crossings (−4 standard deviations from mean signal) and re-sampled at 3× the original rate. Principal component analysis was done on all spikes from each channel, and the first four principal components were used for clustering (*Harris et al., 2000*). A random forest classifier implemented in Python was used to classify spikes according to a model built on a dataset of 1200 clusters manually scored by expert observers. A set of 19 features, including ISI contamination (% of ISIs < 3 ms), similarity to regular spiking unit (RSU) and fast spiking unit (FS) waveform templates, amount of 60 Hz noise contamination and kinetics of the mean waveform. Cluster quality was also ensured by thresholding of L-ratio and isolation distance (*Schmitzer-Torbert et al., 2005*). Clusters were classified as noise, multi-unit

or single-unit. Only single-unit clusters with a clear refractory period were used for FR analysis. We classified units as RSU or FS based on established criteria (mean waveform trough-to-peak and tail slope, *Hengen et al., 2016*). Only RSUs (putative excitatory neurons) were used for analysis. Some units were lost during the recording, presumably due to electrode drift or gliosis. To establish periods of time when units could be clearly detected and isolated, we used ISI contamination: when hourly % of ISIs < 2 ms was above 4%, the unit was considered to be poorly isolated; neurons were included for analysis for each behavioral epoch in which unit isolation quality met criterion. Using these criteria, 93% of cells were well isolated for greater than one-third of the full experiment time, and 80% of cells were well isolated for greater than half of full experiment time of 2–3 days. Results of the automated scoring were manually reviewed for final inclusion of neurons.

## fEPSP analysis

For fEPSP analysis, the data were bandpass filtered (high-pass = 5 Hz, low-pass = 450, filter order = 2). For each optogenetic stimulation, a section of data was extracted (100 ms prior to stimulation and 80 ms after) for each channel in the array. While all the channels were used for estimation of cortical layer placement, only channels in L4 to upper L5 where the response was largest and shortest latency, and channels where signal-to-noise was stable, were chosen for fEPSP analysis. Evoked events were rejected if the standard deviation was >200 μV in the last 5 ms before stimulation or if an evoked trough of >30 μV magnitude could not be found. Slopes were calculated for the rise of the waveform between 20% and 80% of the maximum amplitude. For the sandwich analysis (*Figure 7*), flanking periods of 1 hr before and after a given epoch were compared. Normalized change in amplitude was calculated as post/pre – 1. For time course analysis (*Figure 6*), fEPSP amplitudes were normalized to the first hour of a given epoch. For analysis of the rare epochs that spanned L-D transitions, we normalized the amplitude just after the transition to the amplitude just before the transition to account for L-D mediated changes in amplitude; results were not different if instead these epochs were excluded or were used without this normalization.

## Evoked firing rate analysis

For the evoked spike analysis (*Figure 8*), spike responses were determined for each RSU for each stimulation. Spike responses were measured as the number of spikes that occurred in the 20 ms interval post-stimulation above what was expected given the previous baseline firing rate. A baseline firing rate was determined in the 60 s prior to each stimulation for each cell, and the expected number of spikes given the baseline firing rate was subtracted from the actual spike number that occurred in the 20 ms post-interval: Response = Spike Count − (baseline FR * 0.02 s). To determine whether evoked spikes were above chance levels for a given cell, we performed a bootstrap analysis by randomly sampling times where no stimulations occurred and performed the same analysis. This produced a null distribution of 'responses' to no stimulation. Cells that had responses <95th percentile of randomly sampled responses were considered unresponsive to stimulation and excluded from analysis.

## Statistical analysis

All data analysis was performed using in-house scripts written in MATLAB. For each experiment, means ± SEM derived from individual cell measurements are provided within the Results section of the text, and n's (number of cells and animals), p-values, and statistical tests are provided within figure captions. Generally, a Kruskal-Wallis test was performed on two to three condition experiments in *Figure 3* as the distributions did not pass normality tests. Two-sample Kolmogorov-Smirnov test with a Bonferroni correction for multiple comparisons was used for comparisons between distributions. Wilcoxon signed-rank test was used to compare distributions that were not normally distributed in *Figures 7* and *8*. *Figure 8E*'s data was log normal and so a two-sample t-test after log transformation was used.

For slice experiments where expected means and variance were known, target sample sizes were determined beforehand using power analyses, using a detection threshold of ~10%. For our in vivo experiments, this information was not known so an a priori power analysis was not possible. A post hoc calculation based on sample size and variance shows that changes of >10% would have been detectable.

## Code and data availability

Analysis code is available here: https://github.com/BrianAndCary/papers. (*Cary, 2020*; copy archived at swh:1:rev:ca5ade243dec9e3f5c3cb308eb84f6fcd7be088f

Data is available here: https://figshare.com/projects/Cary_et_al_2021_Elife_Submission/95867.

## Acknowledgements

We thank Alejandro Torrado Pacheco for his invaluable help with the surgery and implementation of in vivo recording and expertise in handling extracellular data and the programming involved in behavioral analysis. We thank Raul Ramos for his technical expertise and aid in imaging of brain slices. Supported by NIH EY025613 (GGT).

## Additional information

### Funding

| Funder | Grant reference number | Author |
| --- | --- | --- |
| National Eye Institute | EY025613 | Gina G Turrigiano |

The funders had no role in study design, data collection and interpretation, or the decision to submit the work for publication.

### Author contributions

Brian A Cary, Conceptualization, Data curation, Software, Formal analysis, Validation, Investigation, Visualization, Methodology, Writing - original draft, Project administration, Writing - review and editing; Gina G Turrigiano, Conceptualization, Resources, Supervision, Funding acquisition, Writing - original draft, Project administration, Writing - review and editing

### Author ORCIDs

Brian A Cary https://orcid.org/0000-0002-1759-164X
Gina G Turrigiano https://orcid.org/0000-0002-4476-4059

### Ethics

Animal experimentation: This study was performed in strict accordance with the recommendations in the Guide for the Care and Use of Laboratory Animals of the National Institutes of Health. All animals were housed, cared for, surgerized, and sacrificed in accordance with Brandeis IBC and IACAUC protocols (#15005 and #18002). All surgery was performed under isoflurane anesthesia. All surgerized animals received two days of post-operative care including daily injection of Meloxicam and Penicillin to reduce discomfort/inflammation and risk of infection. Rats were always housed and recorded with at least one littermate.

### Decision letter and Author response

Decision letter https://doi.org/10.7554/eLife.66304.sa1
Author response https://doi.org/10.7554/eLife.66304.sa2

## Additional files

### Supplementary files

• Transparent reporting form

### Data availability

Processed datasets and all figure data have been uploaded to Figshare (https://figshare.com/projects/Cary_et_al_2021_Elife_Submission/95867).

The following dataset was generated:

| Author(s) | Year | Dataset title | Dataset URL | Database and Identifier |
|---|---|---|---|---|
| Cary BA, Turrigiano GG | 2021 | Cary et. al Elife Submission 2021 | https://figshare.com/projects/Cary_et_al_2021_Elife_Submission/95867 | Figshare, figshare.com/projects/Cary_et_al_2021_Elife_Submission/95867 |

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
