## [Decision Letter]

**Acceptance summary:**

The authors carefully document sleep-wake behavior of young rats during the critical period for development of vision. Building on this, they use an innovative multimodal approach to study synaptic strength in relation to recent sleep-wake behaviors. Of note, they fail to find any relationship between sleep history and the state of excitatory synapses. These results challenge the synaptic homeostasis hypothesis, especially the tenet that sleep universally drives widespread downscaling of synaptic weights.

**Decision letter after peer review:**

[Editors’ note: the authors submitted for reconsideration following the decision after peer review. What follows is the decision letter after the first round of review.]

Thank you for submitting your work entitled "Stability of cortical synapses across sleep and wake" for consideration by *eLife*. Your article has been reviewed by 3 peer reviewers, and the evaluation has been overseen by a Reviewing Editor and a Senior Editor. The following individual involved in review of your submission has agreed to reveal their identity: Anita Lüthi (Reviewer #3).

Our decision has been reached after extensive consultation between the reviewers. Based on these discussions and the individual reviews below, we regret to inform you that your work will not be considered further for publication in *eLife*.

Notably, this report documents surprisingly little evidence for sleep related changes in cortical synaptic plasticity, and therefore challenges the SHY hypothesis. The editors and reviewers appreciate the novel methodology of sleep state detection, careful synaptic analysis with miniature excitatory postsynaptic current recordings and in vivo multi-electrode array local field potential recordings. However, there are concerns about the difficulty in comparing the results of this study to others, given that the approach for picking sleep-related epochs for analysis may be problematic. This concern includes, but is not limited to the result that the sleep dense and wake dense periods appear to be as brief as a few hours (Figure 3A), which may be an insufficient duration to allow significant plasticity.

Given the significant strengths of the end-point analysis (timed recordings of minis and in vivo LFP), if you are interested, we would encourage submission of new manuscript that addresses with new experiments the issues of timing of sleep-related experimental epochs to allow for a more direct comparison to previous studies.

*Reviewer #1:*

The study by Cary and Turrigiano aims to test further the hypothesis that wakefulness is associated with increased synaptic strength and sleep leads to decreased synaptic efficacy. Previous studies on this topic differ widely with respect to the choice of experimental models and methodology, which prevents straightforward comparison between studies. To my opinion the current work is inconclusive as it stands, because it uses experimental models, approaches and data analyses not conventionally used in studies addressing sleep regulatory mechanisms, and does not take into account factors that have important influence on sleep dynamics, behaviour and brain activity.

1. In this study young long-Evans rats were used between postnatal day P25 and 31. Although literature on the development of sleep regulation at this age is limited, evidence suggests that sleep in young rats of a comparable age is more fragmented and a clear cut homeostatic dynamics of SWA is not yet well pronounced: https://journals.physiology.org/doi/abs/10.1152/ajpregu.1990.258.3.R634.

To this end, the first type of analyses I would have suggested is to plot individual hypnograms (as on Figures 2 and 5 in the paper cited above) and identify time points which are not simply preceded by wake-rich or sleep-rich intervals, but differ with respect to homeostatic sleep pressure. Generally, I felt the question asked by the authors could have been better addressed if the experiments were designed and data analyses were performed taking into account the context of daily sleep physiology.

2. It is an important limitation of the study that the animals were generally allowed only a few days of recovery after invasive surgery ("Animals were given 2-3 days of recovery before data collection"). Post op treatment is not mentioned but I would have expected that the animals receive some sort of analgesia for at least a few days as I presume is required by the office of Laboratory Animal Welfare. I would not be surprised if the recovery process or post-op treatment have a significant influence on sleep regulation and on some of the measures taken in this study.

3. The description of the surgical procedure is not entirely clear. It is stated that "For animals to undergo acute slice physiology, we implanted 3 EEG screws" but "For in vivo recording and stimulation experiments, a 16 channel silicon probe" was used. Did you record EEG also in the in vivo recording and stimulation experiments? If I understand correctly, the EEG screws were implanted in the frontal and parietal derivation, and the third screw was a reference above cerebellum. Have you based your "behavioral state classification" on EEG power from the frontal or from the parietal EEG?

4. The procedure for "behavioral state classification" is not very clear. It mentions power bands and their ratios but their choice is not justified. Please provide full absolute EEG power spectra. Please explain how "Large deflections in the EMG" were normalized. It is stated "All classifications made in real time can be manually verified post hoc." Please confirm whether all classifications were actually verified.

Further to this point, it is stated "This real time behavioral state classifier greatly improved our ability to explore the effects of sleep on synaptic strengths, because rodent sleep is both highly variable and fragmented (Figure 1A; see standard deviation Figure 2A)." What exactly is meant by variability here and I do not understand how did your classifier help to deal with that in the context of the question being addressed here?

5. The choice of wake-dense and sleep-dense epochs is, to my view, suboptimal. As well known, the animals may and indeed do spend extended periods of time in a relatively superficial sleep state, where I would not expect any systematic homeostatic changes to occur. The analyses the authors are performing would have been more informative if time intervals compared differ with respect to levels of sleep pressure.

6. For brain slice preparation, the choice of the visual cortex and prefrontal cortex is not well explained. When selecting brain areas for this study I would have looked for evidence of sleep homeostatic response and plasticity at this early age. It would strengthen the argument if such evidence is cited.

7. The approach to detect mEPSC events is not well described. It refers to "Multiple tunable parameters" that were "optimized", but not clear how this was done.

8. This sentence is not clear: "To establish "on" and "off" times for neurons, we used ISI contamination: when hourly % of ISIs < 2 msec was above 4%, unit was considered to be offline." Please clarify what was the purpose of this analysis.

9. Figure 1: Please clarify how the spectrogram was normalized.

10. It is stated that "…on average, the chance of being in a sleep dense epoch between ZT 0-4 was less than 50% (Figure 2B), indicating that circadian time is not a good predictor of sleep history for individual animals." And then later "We wondered whether circadian time might impact postsynaptic strengths independently of sleep/wake history. To test this, we entrained animals to an inverted light/dark cycle…". It is unclear how inverting the LD cycle helps the problem of the potential circadian influence on synaptic strength or any other variable. Does Figure 3A, where ZT0-12 are shaded, shows an example of an inverted LD cycle?

11. It is observed that "mEPSC amplitudes in L2/3 pyramidal neurons were identical after wake dense or sleep dense epochs". Please provide further information on how wake and sleep dense epochs were distributed across the light and the dark periods.

12. Given the pronounced effect of behavioural state on the evoked potentials, it is essential to ensure that the behavioural state is identical for all comparisons. Without knowing the behavioural state it is difficult to conclude whether the effects or lack thereof are related to fluctuations in arousal, movement or reflect changes in synaptic efficacy. I should point out that the figure 4F shows that most longest waking periods are associated with an increase in evoked responses from sleep before to sleep after (please compare hr9 vs hr13, hr34 vs hr38, hr57 vs hr62 or hr82 vs hr86).

*Reviewer #2:*

Cary and Turrigiano conclude from a series of experiments using minis, evoked responses, and firing rates that they are unable to replicate several findings previously reported by the Tononi and Cirelli group and several other laboratories in support of the synaptic homeostasis hypothesis. There are many reasons for these supposedly negative findings. Without conducting a more careful analysis using restrictive criteria to define sustained periods of sleep and waking, and without additional experiments to control for crucial factors such as level of arousal and brain temperature, these results are impossible to interpret. In short, before the authors can conclude that they are unable to replicate the findings by Vyazovskiy et al., and other labs, should conduct the experiments using the same, or similar, carefully controlled conditions.

Minis:

1) There are several reasons that can explain why the authors failed to see significant differences between sleep and waking; (1) the criteria to define sleep dense and wake dense periods (65% of total time for 4 consecutive hours) are not very restrictive; for instance, in Vyazovskiy et al., 2009 spontaneously awake rats were sacrificed during the dark phase after a long period of continuous waking (1 hour, interrupted by periods of sleep not longer than 4 min), and after spending at least 75 % of the previous 6 hours awake; similar criteria were used in de Vivo et al., 2017 in mice; Liu et al., used a lower cut-off, of 65%, but crucially, the spontaneous waking episodes considered for the study occurred during the first part of the dark phase, when mice and rats are much more active; here, mice were studied in the first part of the light period, and as expected and shown in Figure 2B, the probability of single animals to be awake > 65% for 4 consecutive hours was very close to zero; in fact, given figure 2A and B, it is difficult to understand how such long episodes of sustained wake could be found, how many of them, and in how many mice; a figure showing the raw sleep/wake data for all the mice used for the in vitro study (figure 3) should be shown; in line 117, the authors refer to figure 1B, but that figure only shows an example for 60 minutes, not 4 consecutive hours; again, based on figure 2, it is hard to imagine how the authors could find enough episodes in enough mice to perform the experiments; this is further confirmed by the 4 traces shown in figure 3A: none of the 4 examples show >65% waking time for the last consecutive hours; the same issue applies to the layer 2/3 results; also, the legend states that only 3 rats were used for the sleep group, although 4 traces are shown in figure 3A; none of the traces for the inverted wake group are shown, and they should, since results for the inverted waking group actually do show some changes (see below).

2) There are inconsistencies between the data presented by the authors and their conclusions. Firstly, in Figure 3G the authors did find an increased mEPSC amplitude in inverted wake vs sleep animals, however, they downplayed this point in the text by saying "minor shift"; In fact, it was significant according to K-S test (see Figure 3 G legend). The K-S test but not ANOVA is an appropriate statistical test used for examining the amplitude of mEPSCs, because the distribution of mEPSC amplitude is not normalized and the comparison among means from experimental groups is not appropriate. Secondly, In Figure S1F, the authors did show a very significant shift to the left in the cumulative distribution of mEPSC inter-event interval in the WD group as compared with the sleep group in L2/3 PFC neurons (SD-WD p<1e-5), which suggested a higher mEPSC frequency in the WD group than in the sleep group if measured with this parameter. In terms of absolute value, the mEPSC frequency was also higher in WD group than in the sleep group (although it was not significant). Therefore, the statement that synaptic strength was stable across sleep/wake periods is questionable at least in these L2/3 PFC neurons.

3) It is not clear whether the recording of mEPSCs from naturally wake and sleep rats was well controlled throughout the investigations. It seems that the preparation of slices from these two groups were performed at different times of the day. This means slices were cut at different time points for these two groups (Figure 3A is misleading). Therefore, the variation in slice conditions may mask the difference between groups. Although the authors did sample a big number of cells for each group, I am not sure whether this will help to limit the effect of variation resulting from the slice preparations. Note that Liu et al., took care of running paired experiments, in which one slice from a control animal and one from a waking/sleep deprived animal were always run in parallel the same day, to limit technical variability.

4) The rationale for selecting layer 4 should be better justified (line 124); firing rates vary across waking and NREM sleep across the entire thalamocortical system, not just in layer 4; thus, taken alone this is not a compelling reason to select layer 4 neurons; on the other hand, it is well known that after the end of the critical period, the thalamocortical synapses targeting layer 4 of primary somatosensory, auditory, and visual cortex lose most of their ability to undergo plastic changes under physiological conditions, and that ability can be reinstated only by specific manipulations such as prolonged unimodal or crossmodal sensory deprivation or peripheral nerve transection. Thus, it seems that the authors of this study chose to focus on synapses that are known to have little plasticity to test synaptic homeostasis hypothesis, whose main claim is that sleep is the price for plasticity during waking; in fact, if the results related to layer 4 could be trusted (but see all the issues related to selection of behavioral states and minis analysis), then they would actually be a nice confirmation of the main tenet of this hypothesis.

Evoked responses:

The analysis of the evoked responses is impossible to interpret because too many crucial details and control experiments were not performed.

1) First, the criteria to define prolonged periods of sleep and waking for the evoked responses analysis are not specified and cannot be deduced from Figure 5A, which has no time bar (same problem in Figure 6). In Vyazovskiy et al., a decrease in slope was present only after at least 2 hours of consolidated sleep, or more than one hour of continuous waking (most rats were awake for 2-4 hours). Vyazovskiy also stimulated only twice, before and after sleep or waking, while it seems that in the current study pulses were given every 20 to 40 secs continuously, for days.

2) Second, evoked responses are exquisitely sensitive to neuromodulatory conditions (arousal levels) and subtle changes in arousal could mask any subtle effect due to sleep/waking history. Vyazovskiy et al. took great care in controlling for this factor by delivering the stimuli under a very standardized quiet waking condition, which required 2 investigators watching the animal full time. As they state, "We did not attempt to record evoked responses continuously for several hours in freely behaving rats because it is impossible to maintain the animals in a standard quiet wakefulness for more than a few minutes." Moreover, Vyazovskiy et al. confirmed that changes in slope were present after controlling for response amplitude. Note that their major results were confirmed by comparing high vs low sleep pressure in all 4 behavioral states separately.

3) The classifier distinguished 3 states, but not active and quiet waking (line 87). This is a crucial limitation because waking responses vary due to arousal levels (see point 2), and differ between quiet and active waking. There is strong evidence from electrophysiological and calcium imaging data that the activity of V1 neurons is very sensitive to locomotion; thus pooling evoked responses across "waking" is inappropriate.

4) Third, evoked responses are exquisitely sensitive to brain temperature. Very small changes in brain temperature can affect evoked responses and mask any additional effect due to sleep and waking history. Vyazovskiy et al. controlled for this factor by conducting specific experiments in which brain temperature was also measured; in doing so, they could demonstrate that the changes in the slope of the evoked response did not correlate with changes in brain temperature. This issue is especially crucial in the current study, where light pulses were used to evoke the response. On a related matter, the intensity of the laser stimulation should be specified.

Firing rates:

The current negative findings relative to firing rates in V1 are at odds with the evidence provided by at least 3 different labs showing that mean firing rates decreases with sleep, including Vyazovskiy et al., in barrel cortex (Nature 2011), Grosmark et al., in the hippocampus (Neuron 2012), Watson et al., in frontal cortex (Neuron 2016), Miyawaki et al., in the hippocampus (Curr Biol 2016, Cell Reports 2019). As for the evoked responses, it is unclear whether the criteria used by Cary and Turrigiano to define prolonged periods of sleep and waking were stringent enough to match those used in other studies. Cary and Turrigiano cite one paper from their lab (line 57) showing that mean firing rates do not change during extended periods of sleep and waking. At the very least, it would be appropriate to quote all the other studies that found the opposite.

The authors state that many units were lost in the course of the several days of recordings. The exact number should be stated.

*Reviewer #3:*

General assessment. This paper asks how states of sleep and wakefulness regulate global synaptic strength at various cortical pyramidal neurons. A major proposition for this question is formulated in the well-known SHY hypothesis, for which there is mostly indirect molecular and structural evidence. Therefore, it is very important to test SHY with direct functional measures of synaptic strength. This paper does so using electrophysiological methods and is, therefore, an important contribution to a long overdue question.

The authors depart from a form of homeostatic plasticity that is known to be regulated by sensory experience and largely based on amplitude measurements of mEPSCs. When now applied to sleep and wakefulness, results are overall negative, thus questioning that SHY affects homeostatic plasticity. The experiments are well-done and the results are clear and striking.

Still, I would encourage the authors to consider a number of points in a revised version of their manuscript.

1) Insufficient information about animal husbandry is provided. All experiments are done in young rats around and shortly after weaning. Weaning changes metabolism and stress levels are high. Synapse growth and development progress rapidly. When were animals weaned relative to the day of surgery? How were they housed prior to and after surgery, and how was recovery from surgery monitored (weight loss and recovery, stress monitoring, etc.)? A time period of 2-3 days for recovery from surgery is very short (~1 week is typical). How much time was given for habituation to the tethering to the recording cables (~1 week is typical)? Is the sleep-wake behavior of the animals stable from 2-3 days after surgery?

2) More information on the sleep-wake behavior of these young rodents is also needed. Figure 2 suggests that the typical preference for sleep over wake during the light period found in adult is not there yet. Do these animals show homeostatic regulation of sleep? The SHY hypothesis implies slow-wave activity in the renormalization of synaptic strength. Slow-wave activity is proposed to be key for synaptic scaling during sleep. Therefore, it would be important to show some evidence that slow-wave activity varies with time-of-day and/or after sleep deprivation.

3) Experiments coincide with critical periods of the visual system. If animals are taken 7-10 days later, after the closure of the critical period, are effects of prior sleep-wake history still negative? What about thalamocortical projections for which the critical period closed much earlier, such as for the whisker-to-barrel system?

4) Regarding the sleep-wake monitoring and analyses, a major weak point is that scoring of vigilance states is done in 10 sec intervals, which is 2.5x the window commonly used. This means that the duration of NREMS bouts are overestimated because brief arousals will go undetected. There should be an estimation provided for the limited time resolution.

5) The authors only show "Time awake" prior to slice preparation. I suggest they instead show time spent in NREM and REM sleep prior to sacrifice. Possibly, then, animals should be selected based on how much time they spent in NREM only.

6) Major forms of homeostatic synaptic plasticity in sensory cortices (e.g. effects of monocular deprivation) develop over time scales of days. The authors show themselves in a previous work that after a full day of MD, mEPSC amplitude is reduced to only ~95% of control (Lambo, 2013). Similarly, in cultures, homeostatic scaling of excitatory or inhibitory synapses is typically observed after 24 h of receptor antagonism. In contrast, here, authors work with time intervals of 4h during which both sleep and wake are present, with mean bout length on the order of hundreds of seconds. This could mean that changes in mEPSC amplitude might simply not be detectable at this point. The mEPSC as a measure for synaptic strength could thus not be sensitive enough. The same considerations might apply for prefrontal cortex, for which the maturational profile is even less known.

7) More generally, one might wonder about whether mEPSCs are suitable for monitoring changes in synaptic strength. Miniature EPSCs reflect the response of a synapse to the release of a single vesicle. Thalamocortical synapses, however, activate their postsynaptic targets via multivesicular release. This might lead to more vigorous postsynaptic receptor recruitment. Therefore, to fully assess whether or not sleep-wake modify synaptic strength, it would be important to look at how synaptic strength quantified by action-potential-dependent vesicular release is affected. I suggest to use single-fiber stimulation to trigger vesicular release via single action potentials.

8) Looking at evoked fEPSPs in response to sensory or afferent stimulation has a long tradition but is not a very informative type of data. Evoked fEPSPs are not only composed of sources arising from the synaptic input, but also by the tendency of the network to switch between up and down states. This is particularly the case during nREM sleep because on-going oscillations are strong and excitatory input can switch the networks between states. So, neither amplitude nor slope of these responses, nor their stability across sleep periods, tell much about "synaptic strength".

[Editors’ note: further revisions were suggested prior to acceptance, as described below.]

Thank you for resubmitting your work entitled "Stability of neocortical synapses across sleep and wake" for further consideration by *eLife*. Your revised article has been evaluated by John Huguenard (Senior Editor).

As you will see from the reviewers comments below, the manuscript has been improved but there are remaining issues that need to be addressed, as outlined below. Normally these would require additional experiments, but given pandemic conditions that limit feasibility for such, then at the very least the title, abstract and conclusions need to be moderated to reflect that the conclusions may be limited to an early developmental period. Further, upon careful inspection of the in vivo LFP data, I see that there is a potential confound that needs to be examined. While these are clearly thalamic-dependent LFP responses, as they are recorded in the cortex, and evoked by ChR2 activation of thalamocortical projections, what is not clear is how one might distinguish the specific thalamic fEPSP from the overall LFP response especially with optogenetic stimulation which can lead to a significant fiber volley (see PMID: 27489370).

*Reviewer #1:*

They authors start with a comprehensive documentation of sleep-wake behavior of young rats during the critical period for vision, followed by an ingenious approach to study synaptic strength as a function of spontaneous recent sleep-wake behaviors, followed by an in-depth analysis of in vitro and in vivo correlates of synaptic strength. I particularly liked the explicit way of the authors in motivating their choice of parameters regarding sleep-wake behavior, of animals during the critical period, and of the synapses studied in vitro or in vivo.

The authors carefully conclude that their results do not support the idea that sleep or wakefulness per se lead to global modifications of synaptic strength, at least not in visual and prefrontal cortex. Interestingly, however, the authors identified circadian variations specifically in prefrontal cortex, an observation that is worth pursuing in the future.

This manuscript is a long awaited and authoritative approach to challenge the SHY hypothesis with solid experimental quantification of functional synaptic strength.

I have a few additional comments to further improve some analysis and their documentation in this study:

1) The sleep-wake behavior of these young animals is clearly different than the one from adult animals. It is irregular and polyphasic and there is very little light-dark dependence (see Figure 1B). Therefore, it would be good to show hourly mean times spent in the different vigilance states wake, NREM and REM rather than only the 4h-slidingwindow means.

2) Beyond the mean times, the detailed architecture of sleep-wake behavior in the 4h-windows is also important. Why is this: even if you go for times enriched in wake or sleep, it is not the same whether this enrichment happens in many very brief bouts or in few relatively long bouts. This is particularly the case for sleep, for which fragmentation has a strong effect on plasticity/learning. Therefore, do the animals sleep in consolidated bouts, i.e. what is the mean NREM sleep bout duration? Is this duration variable between Sleep dense and wake dense, and as a function of circadian time?

3) The word "consolidated" has a strong meaning in the sleep field and as it refers to the mean duration of NREMS bouts – the less they are interrupted by microarousals, the more consolidated NREMS is said to be. It should not be used to describe an enrichment of mean times spent in sleep or wake over a 4-h period (see line 113).

4) Line 116: δ power is not the same as the size of slow waves and should not be equated. Slow waves are EEG or LFP graphoelements that at best make up a fraction in the power of the broad δ frequency band used here (0.5-4Hz). Please use these terms carefully and specify the frequency bands upon first use.

5) Line 119. The time-of-day-dependence of δ power is not typically referred to as an oscillation. Also, the way the data in Figure 1C analyzed should be checked. It must be done for equivalent amounts of time spent in NREM sleep epochs that are preceded and followed by other NREM sleep epochs. If it is not done like this, the amounts of δ power at different times of day are not weighed equivalently. Therefore, please divide the total time spent in NREMS in the light and in the dark phase into similar amounts and calculate δ power within these bouts. As more time is spent in NREMS in the light phase, subdivisions can be higher in the light than in the dark phase (e.g. 12 time points in the light and 6 in the dark phase). Literature from the Paul Franken lab can be consulted to do this properly.

6) Lines 168-172: Please explain more quantitatively. What is a long natural wake epoch during the early light phase? What was the criterion to add a novel object? What you do here strictly amounts to a sleep deprivation that should be documented in terms of its effects on quiet vs mobile wakefulness and the increase in the time spent in wakefulness. One could also argue that this is a period of environmental enrichment that has an impact on its own on visual plasticity. These animals might also show a greater increase in δ power upon the end of wakefulness due to greater amounts of sleep loss. These caveats should be quantified whenever possible and discussed.

7) Figure 3 —figure supplement 1. This figure indicates somewhat worrisomely that the increase in δ power at light onset compared to the end of the light period is extremely variable – from somewhere between >40%, which is very high, to ~17%, which is very low. In addition, the example recording shown in panel A is close to 50%, so where is that datapoint represented in panel B? What happened with these animals during the darkphase that their sleep pressure at light onset is so variable? Overall, I am not sure that this analysis is particularly helpful because it relates an endpoint measure of synaptic amplitudes to an unknown starting point measure.

8) Line 624. What are slice behavioral data?

9) The finding that evoked field potential amplitudes in visual cortex were larger in NREMS than in wakefulness during both the light and the dark phase is intriguing and in contrast with previous observations on evoked auditory field responses (see e.g. the literature from Yaniv Sela et al.,). Rather what has been seen is that the secondary outward components of the evoked responses are disproportionately increased during NREM sleep. Evoked field responses are, as already mentioned in my first review, also problematic because they can be contaminated because of on-going oscillatory activity. Can the authors better describe when these responses were elicited in response to on-going up- and downstates, for example? And how these components were removed to isolate the evoked field response?

*Reviewer #2:*

In the revised manuscript, the authors included important and interesting new analyses, but no new experiments have been undertaken, and the key issue remains that the data set, limited to results obtained in juvenile animals and analyzed in some novel, yet untested ways, is not ideal for reaching strong conclusions.

My main points I raised previously remain the same:

1. The experiments have been performed in juvenile animals, which prevents direct comparisons with most previous studies. I recommend that the title of the manuscript should state the age of animals to avoid misunderstanding.

2. The experiments were performed too soon after a major invasive surgical procedure, which is certainly expected to affect sleep and wake quality, because the immune system and thermoregulation can be still compromised especially in young animals.

3. The overall wake-sleep pattern is too fragmented for a straightforward comparison between effects of wake and sleep, and I still do not see evidence provided that homeostatic sleep pressure is increased during time periods referred to as "wake-dense" epochs, and decreases during "sleep-dense" epochs.

4. The chronic fEPSP experiment shows massive state- and light-dark-dependent instantaneous variations in the amplitude of responses, which makes it very difficult to extract any meaningful sleep-wake history dependent changes, and the potential direct influence of arousal, behaviour and brain temperature on evoked responses remains not addressed.

[Editors’ note: further revisions were suggested prior to acceptance, as described below.]

Thank you for resubmitting your work entitled "Stability of critical-period neocortical synapses across sleep and wake states" for further consideration by *eLife*. Your revised article has been evaluated by John Huguenard (Senior Editor) and a Reviewing Editor.

The manuscript has been improved but there are some remaining issues that need to be addressed, as outlined below:

We think this is an important paper providing experimental data regarding the sleep homeostatic hypothesis using an innovative approach. Please consider the following comments from reviewers in your final revision of this paper. We will not need to send your revision back to reviewers. Instead the reviewing editor will make the final call.

*Reviewer #1:*

The authors did an excellent job replying to my concerns. In my view, the study now makes abundantly clear that it studies critical period synapses. The study also has appropriately addressed remaining analytical concerns, with one exception:

This concerns the re-analysis of the data presented in the Reviewer Figure 1. I appreciate that the authors re-binned their data according to the light-and dark phase. However, this analysis is NOT about simple binning. It is about calculating δ power for similar times spent in NREMsleep. Therefore, it is about dividing the total amount of time spent in NREMS in different percentiles in the light and dark phase. Accordingly, the datapoints will not be placed in regular time intervals, but they will be displaced according to where the mean values of the time bin come to lie.

While I can understand that this can be considered as a detail, I encourage the authors to present their main figure according to the standards in the field.

My second remaining concern is that the authors should once again read the recommendations of the Senior Editor to use careful wording regarding the evidence against SHY given all that was discussed. For example, I still see the last sentence of the abstract "…strong evidence against the view that sleep drives widespread downscaling" unchanged.

Finally, the Supplement to Figure 5 seems not necessary to me.

*Reviewer #2:*

The change in the title is welcome, but no attempt has been made to address my other comments. My original comments were:

– The experiments were performed too soon after a major invasive surgical procedure, which is certainly expected to affect sleep and wake quality, because the immune system and thermoregulation can be still compromised especially in young animals.

The current version of the manuscript states, for example, "Animals were given 2-3 days of recovery before data collection, at which point sleep/wake behavior had stabilized and was comparable to unsurgerized animals." No data were provided to support this statement, and in my view this is unacceptable for any study that attempts to address physiological functions of sleep.

The overall wake-sleep pattern is too fragmented for a straightforward comparison between effects of wake and sleep, and I still do not see evidence provided that homeostatic sleep pressure is increased during time periods referred to as "wake-dense" epochs, and decreases during "sleep-dense" epochs.

I would like to refer the authors to extensive literature which describes modelling of sleep homeostasis (based on δ power, or firing rates), specifically studies that show individual examples of how Process S changes as a function of sleep-wake states. From those studies it is clear that the dynamics occuring as a function of sleep-wake states are expected to be relatively slow, and one cannot expect prominent changes during wake or sleep states unless these are consolidated.

– The chronic fEPSP experiment shows massive state- and light-dark-dependent instantaneous variations in the amplitude of responses, which makes it very difficult to extract any meaningful sleep-wake history dependent changes, and the potential direct influence of arousal, behaviour and brain temperature on evoked responses remains not addressed.

I do not have anything to add to this point, except I would like to emphasise that unless the major light-dark differences in the amplitude of evoked responses are explained, it cannot be considered a reliable measure of synaptic strength.

Altogether, I do not think this study provides data that support conclusions such as "our data show that synaptic downscaling is not a universal function of natural sleep across cortical cell types and brain regions".

---

## [Author Response]

[Editors’ note: the authors resubmitted a revised version of the paper for consideration. What follows is the authors’ response to the first round of review.]

Notably, this report documents surprisingly little evidence for sleep related changes in cortical synaptic plasticity, and therefore challenges the SHY hypothesis. The editors and reviewers appreciate the novel methodology of sleep state detection, careful synaptic analysis with miniature excitatory postsynaptic current recordings and in vivo multi-electrode array local field potential recordings. However, there are concerns about the difficulty in comparing the results of this study to others, given that the approach for picking sleep-related epochs for analysis may be problematic. This concern includes, but is not limited to the result that the sleep dense and wake dense periods appear to be as brief as a few hours (Figure 3A), which may be an insufficient duration to allow significant plasticity.Given the significant strengths of the end-point analysis (timed recordings of minis and in vivo LFP), if you are interested, we would encourage submission of new manuscript that addresses with new experiments the issues of timing of sleep-related experimental epochs to allow for a more direct comparison to previous studies.

We have significantly extended our data and analyses to address the major points raised above, and have also performed extensive textual revisions to clarify our experimental logic and procedures, and to explain how our paradigms and results compare to previous studies.

The major changes to the manuscript are:

1. To address the timing and characteristics of sleep related epochs, we have performed a detailed analysis of the sleep patterns of the young (P25-31) Long-Evans rats used in these experiments. Consistent with previous reports we find modulation of the amplitude of slow wave sleep (SWA) at this age (new Figure 1C), and in general their sleep/wake amount and distribution between light and dark phases are quite similar to those reported previously for older Long-Evans rats (Figure 1; Frank and Heller, 1997; Frank et al., 2017).

2. We have added data showing that even very long periods of natural sleep and wake – up to 5 hr (sleep dense) or 13 hr (wake dense) – do not drive changes in thalamocortical synaptic efficacy in neocortical neurons. These data are presented in the new Figures 6-8. We note that the longest epochs we analyze are as long as these animals will naturally experience, so if an important function of sleep is to constitutively drive synaptic changes then they should occur over this time frame.

3. We provide additional analysis of our mEPSC data. We show that in addition to mEPSC amplitude being stable after sleep and wake dense epochs, this measure of postsynaptic strength does not correlate with the proportion of time spent awake during wake dense epochs (Figure 3 figure supplement 1E), or with the drop in δ power during sleep dense epochs (Figure 3 figure supplement 1B).

4. We note that the criteria we use for sleep and wake dense epochs, the age range of our animals, and the brain regions examined are comparable or overlapping with those used in a range of earlier studies (summarized in the attached reviewer Table 1 and now discussed in detail in the text).

5. We now extend our behavioral state classification to include REM, NREM, and active and quiet wake (Figure 2 —figure supplement 1), and quantify changes in fEPSP amplitude and evoked spikes separately for each of these 4 states (new Figures 6-8). Further, we find that L-D transitions rapidly modulate fEPSP amplitude, and we now account for this in our analysis (new Figure 5G).

6. We performed a number of additional analyses and extended and revised most of the figures to better support our major conclusions; these are explained in our point-by-point response to the reviewers below.

7. We have completely re-written the manuscript to better motivate and explain our approach and to incorporate these new data and analyses, and in particular we now strive to make clear how our data fit in with previous findings.

Reviewer #1:The study by Cary and Turrigiano aims to test further the hypothesis that wakefulness is associated with increased synaptic strength and sleep leads to decreased synaptic efficacy. Previous studies on this topic differ widely with respect to the choice of experimental models and methodology, which prevents straightforward comparison between studies. To my opinion the current work is inconclusive as it stands, because it uses experimental models, approaches and data analyses not conventionally used in studies addressing sleep regulatory mechanisms, and does not take into account factors that have important influence on sleep dynamics, behaviour and brain activity.

We respectfully disagree that our approach differs fundamentally from those used previously. Previous studies have used (1) changes in mEPSC amplitude/frequency (Liu et al., 2010; Khlghatyan et al., 2020), (2) changes in firing rates (Vyazovskiy et al., 2009; Miyawaki and Diba, 2016), (3) changes in evoked fEPSPs (Vyazovskiy et al., 2008), and (4) changes in synapse morphology or protein composition (de Vivo et al., 2017; Diering et al., 2017), to argue that sleep and wake drive opposite changes in synaptic strengths; here we use 3 of these 4 standard approaches in the same study. The major difference between our study and previous studies is that here we combined a careful analysis of the wake and sleep history of each animal with rigorous measurements of each of these correlates of synaptic efficacy. We appreciate the reviewer’s point that we originally did not provide some important details of our sleep/wake analysis, and we have now revised the manuscript to include these details, as described in our response to specific points below.

1. In this study young long-Evans rats were used between postnatal day P25 and 31. Although literature on the development of sleep regulation at this age is limited, evidence suggests that sleep in young rats of a comparable age is more fragmented and a clear cut homeostatic dynamics of SWA is not yet well pronounced: https://journals.physiology.org/doi/abs/10.1152/ajpregu.1990.258.3.R634.To this end, the first type of analyses I would have suggested is to plot individual hypnograms (as on Figures2 and 5 in the paper cited above) and identify time points which are not simply preceded by wake-rich or sleep-rich intervals, but differ with respect to homeostatic sleep pressure. Generally, I felt the question asked by the authors could have been better addressed if the experiments were designed and data analyses were performed taking into account the context of daily sleep physiology.

Our experimental paradigm was explicitly designed to take circadian aspects of sleep physiology into account, but we did not make this point clearly in our original presentation of the data. As suggested, we now provide many hypnogram examples, and sleep histories for each animal (Figure 3, Figure 3 – supplement 2; Author response images 1-5), as well as an analysis of slow wave activity (SWA) across the circadian cycle (new Figure 1C). These show that LE rats of this age have sleep patterns that look similar to older animals: they sleep more during the light phase as expected, and show oscillations in SWA that are higher at the beginning of the light phase (when they have slept less) and decrease as they accumulate sleep, then rise again during the dark phase as they spend more time awake (Figure 1C). We agree these data are important for interpreting our results, and we now show that sleep-dense periods generally ensue when SWA is high, as expected. Further, we now provide an analysis showing that there is no correlation between the drop in sleep pressure (as assessed by slow wave power) during a sleep dense epoch, and mEPSC amplitude at the end of that epoch (new Figure 3 —figure supplement 1).

**Author response image 1. sa2fig1:** Sleep history from start of EEG recording for animals used for slice experiments. Unperturbed sleep patterns from initiation of EEG recordings, which started 2-3 days postsurgery. Dots show proportion of time spent awake for individual animals (averaged in 1 hr bins). Green line shows average across animals, shading = SEM. Sleep/wake behavior stabilized soon (roughly 4-8 hr) after the initiation of recording..

**Author response image 2. sa2fig2:** Endpoint behavioral state proportions. Top row (Last 4 hrs), represents the relative time spent in each behavioral state in the last 4 hrs prior to sacrifice for each individual animal used for *ex vivo* slice recording. The top row is separated in to three columns for each type of experiment. From left to right, sleep dense experiments (in light period), light period wake dense experiments, dark period wake dense experiments. Green and blue dotted lines represent wake (>65%) and sleep (<35%) dense thresholds, respectively. Bottom row (Last hour), is identical to the top row but for the last hours of time prior to sacrifice. Black bars represent mean and 95% confidence interval.

**Author response image 3. sa2fig3:** Visual Cortex L4 hypnograms. Here the hypnograms for each individual animal used in L4 visual cortex *ex vivo* slice experiments are plotted. Green bars indicate instances of wake, magenta indicates REM, and blue NREM. Grey rectangle indicates hours when lights are off (i.e. the dark period). Top plot, hypnograms for animals in the light period sleep dense experiment. Bottom plot, hypnograms from light period wake dense experiment.

**Author response image 4. sa2fig4:** Visual Cortex L2/3 hypnograms. Here the hypnograms for each individual animal used in L2/3 visual cortex *ex vivo* slice experiments are plotted. Green bars indicate instances of wake, magenta indicates REM, and blue NREM. Grey rectangle indicates hours when lights are off (i.e. the dark period). Top plot, hypnograms for animals in the light period sleep dense experiment. Middle plot, hypnograms from light period wake dense experiment. Bottom plot, hypnograms from dark period wake dense experiment.

**Author response image 5. sa2fig5:** Prefrontal Cortex L2/3 hypnograms. Here the hypnograms for each individual animal used in L2/3 prefrontal cortex *ex vivo* slice experiments are plotted. Green bars indicate instances of wake, magenta indicates REM, and blue NREM. Grey rectangle indicates hours when lights are off (i.e. the dark period). Top plot, hypnograms for animals in the light period sleep dense experiment. Middle plot, hypnograms from light period wake dense experiment. Bottom plot, hypnograms from dark period wake dense experiment.

We would also like to point out that studies in support of sleep-dependent regulation of synaptic strength in rodents have spanned a wide age range, including animals of a comparable (e.g. Liu et al., 2010; de Vivo et al., 2017; Spano et al., 2019) or even much younger age (de Vivo et al., 2019); sleep is thought to be especially important in young animals, so if a fundamental role of sleep is to reduce synaptic efficacy, we might expect to see it in young animals during this highly plastic phase of postnatal development.

2. It is an important limitation of the study that the animals were generally allowed only a few days of recovery after invasive surgery ("Animals were given 2-3 days of recovery before data collection"). Post op treatment is not mentioned but I would have expected that the animals receive some sort of analgesia for at least a few days as I presume is required by the office of Laboratory Animal Welfare. I would not be surprised if the recovery process or post-op treatment have a significant influence on sleep regulation and on some of the measures taken in this study.

Our standard protocol for these chronic recordings is to begin habituating animals to the recording chamber on the third day post surgery (e.g. Hengen et al., 2013, 2016; Torrado Pacheco, 2020). Animals are fully recovered at this point (eating and drinking normally, gaining weight, grooming normally, etc.; following NIH guidelines and our IACUC approved protocol), and this timeframe gives us the greatest recording stability for these chronic recordings. We have shown previously using this same surgery and recovery protocol that baseline firing rates are already stable after this recovery period (Hengen et al., 2016; Torrado Pacheco, 2020). We have now verified that sleep patterns in our animals have also stabilized prior to the initiation of experiments (Author response image 1).

3. The description of the surgical procedure is not entirely clear. It is stated that "For animals to undergo acute slice physiology, we implanted 3 EEG screws" but "For in vivo recording and stimulation experiments, a 16 channel silicon probe" was used. Did you record EEG also in the in vivo recording and stimulation experiments? If I understand correctly, the EEG screws were implanted in the frontal and parietal derivation, and the third screw was a reference above cerebellum. Have you based your "behavioral state classification" on EEG power from the frontal or from the parietal EEG?

The methods have been expanded and clarified throughout, and we now make clear that EEG was recorded for the sleep classification in slice experiments, while for the thalamocortical stimulation experiments we used LFPs from the implanted electrodes (Methods, lines 565-566). The EEG signal was recorded as the difference between the parietal and frontal screw, grounded to the cerebellum screw. This has been added to the methods.

4. The procedure for "behavioral state classification" is not very clear. It mentions power bands and their ratios but their choice is not justified. Please provide full absolute EEG power spectra. Please explain how "Large deflections in the EMG" were normalized. It is stated "All classifications made in real time can be manually verified post hoc." Please confirm whether all classifications were actually verified.

We have now re-written the Results section and figure legend to better explain and justify our classification procedure, and we have included additional details on the behavioral recording and classification in the Methods (lines 577-596); yes, all sleep state classifications were verified post-hoc, and we now provide a benchmark of our automatic classification against manual classification (revised Figure 2D).

Further to this point, it is stated "This real time behavioral state classifier greatly improved our ability to explore the effects of sleep on synaptic strengths, because rodent sleep is both highly variable and fragmented (Figure 1A; see standard deviation Figure 2A)." What exactly is meant by variability here and I do not understand how did your classifier help to deal with that in the context of the question being addressed here?

We apologize for not making our reasoning clear. There is individual variability between animals in exactly when they experience sleep dense or wake dense epochs (new Figure 1, compare animals 1 and 2). By classifying sleep in real time we can detect these epochs when they occur, rather than relying on the average behavior, which would result in mis-classification in some animals (new Figure 1). We have bolstered this point by showing individual hypnograms (Figure 3, Figure 3 – supplement 2), as well as examples of individual (Figure 1A) and average (Figure 1B-D) sleep-wake behavior.

5. The choice of wake-dense and sleep-dense epochs is, to my view, suboptimal. As well known, the animals may and indeed do spend extended periods of time in a relatively superficial sleep state, where I would not expect any systematic homeostatic changes to occur. The analyses the authors are performing would have been more informative if time intervals compared differ with respect to levels of sleep pressure.

To disentangle the impact of sleep and wake from circadian time it is necessary to compare sleep dense and wake dense epochs from a similar circadian period, where on average the sleep pressure will be similar; we also analyzed wake dense periods from the opposite circadian time when sleep pressure will on average be different (see new Figure 1, Figure 3). Most studies designed to test SHY have used a similar design. However, for each individual animal the sleep pressure at the beginning of a sleep dense epoch will differ (as it depends on previous sleep history). Therefore, to more directly address the question of whether sleep pressure is predictive of changes in synaptic strength we looked at the relationship between the change in slow wave amplitude across a sleep dense epoch and mEPSC amplitude at the end of the epoch. We found no relationship (new Figure 3 – supplement 1), consistent with the overall lack of impact of sleep on synaptic strengths.

6. For brain slice preparation, the choice of the visual cortex and prefrontal cortex is not well explained. When selecting brain areas for this study I would have looked for evidence of sleep homeostatic response and plasticity at this early age. It would strengthen the argument if such evidence is cited.

Visual cortex was chosen because there is extensive plasticity during the visual system critical period, and we have shown previously that synaptic up and downscaling can be induced at this age in V1 (Hengen et al., 2016; Torrado Pacheco et al., 2020). Prefrontal cortex was chosen because a previous study reported sleep/wake dependent changes in mEPSC amplitude in this brain area, from animals of a comparable age (Lui et al., 2010). We have now made this rationale more explicit in the Results section (lines 220-223).

7. The approach to detect mEPSC events is not well described. It refers to "Multiple tunable parameters" that were "optimized", but not clear how this was done.

We now describe our method clearly in the methods section; the approach we use has been published previously, and allows us to do largely automated detection (Miska et al., 2018; Torrado Pacheco et al., 2020).

8. This sentence is not clear: "To establish "on" and "off" times for neurons, we used ISI contamination: when hourly % of ISIs < 2 msec was above 4%, unit was considered to be offline." Please clarify what was the purpose of this analysis.9. Figure 1: Please clarify how the spectrogram was normalized.

For the chronic evoked-spike recordings we need to determine when during the experiment we have well-isolated units; how these “on” and “off” times are determined is now more clearly explained in the spike extraction, clustering, and sorting subsection of methods (lines 720-726). The spectrogram heatmap color was normalized to the maximum power value, such that all power values for each frequency were represented as a fraction of the highest power in the 1 hour segment of data shown (this has been added to the legend).

10. It is stated that "…on average, the chance of being in a sleep dense epoch between ZT 0-4 was less than 50% (Figure 2B), indicating that circadian time is not a good predictor of sleep history for individual animals." And then later "We wondered whether circadian time might impact postsynaptic strengths independently of sleep/wake history. To test this, we entrained animals to an inverted light/dark cycle…". It is unclear how inverting the LD cycle helps the problem of the potential circadian influence on synaptic strength or any other variable. Does Figure 3A, where ZT0-12 are shaded, shows an example of an inverted LD cycle?

We did not explain our reasoning clearly in the original version of the manuscript. Our purpose was to compare sleep dense and wake dense periods that occurred during the light cycle (within a 5 hr circadian window) to compare the impact of sleep and wake largely independently of circadian time; we also wished to compare wake dense epochs that occurred during opposite circadian times, to probe for any influence of circadian cycle on mEPSC amplitude. We inverted the L-D cycle in this later group of animals in order to keep all other experimental variables (time of slicing and recording) the same for the experimenter. We have now clarified this in the methods section (lines 537-542).

11. It is observed that "mEPSC amplitudes in L2/3 pyramidal neurons were identical after wake dense or sleep dense epochs". Please provide further information on how wake and sleep dense epochs were distributed across the light and the dark periods.

The sleep/wake history and endpoint (including when in the L or D cycle) for each animal for the slice experiments is now shown in Figure 3, and Figure 3 – supplement 2 (Author response images 2-5 for full details). Measurements were taken within two windows: either ZT 3-8 (after sleep or wake dense epochs), or ZT 13-16 (after wake dense epochs).

12. Given the pronounced effect of behavioural state on the evoked potentials, it is essential to ensure that the behavioural state is identical for all comparisons. Without knowing the behavioural state it is difficult to conclude whether the effects or lack thereof are related to fluctuations in arousal, movement or reflect changes in synaptic efficacy. I should point out that the figure 4F shows that most longest waking periods are associated with an increase in evoked responses from sleep before to sleep after (please compare hr9 vs hr13, hr34 vs hr38, hr57 vs hr62 or hr82 vs hr86).

The reviewer raises an excellent set of points that prompted us to undertake a significant additional analysis of the impact of behavioral state and L-D periods on fEPSPs. The increase in evoked responses that the reviewer noticed was actually due to the transitions between L and D (not noted on the original figure). We have now separated our fEPSP measurements by behavioral state and by ZT time (new Figure 58) so that this effect can be clearly seen. For all 4 behavioral state classifications (REM, NREM, active wake and quiet wake) there is a clear and rapid change in fEPSP amplitude at L-D transitions, and no other significant changes as a function of ZT time (new Figure 5G). We now analyze the impact of time asleep or awake on fEPSP amplitude and slope independently for each behavioral state, and take the L-D transition into account; consistent with our earlier data, we find that fEPSPs do not change as a function of time spent awake or asleep (new Figure 6, 7).

Reviewer #2:Cary and Turrigiano conclude from a series of experiments using minis, evoked responses, and firing rates that they are unable to replicate several findings previously reported by the Tononi and Cirelli group and several other laboratories in support of the synaptic homeostasis hypothesis. There are many reasons for these supposedly negative findings. Without conducting a more careful analysis using restrictive criteria to define sustained periods of sleep and waking, and without additional experiments to control for crucial factors such as level of arousal and brain temperature, these results are impossible to interpret. In short, before the authors can conclude that they are unable to replicate the findings by Vyazovskiy et al., and other labs, should conduct the experiments using the same, or similar, carefully controlled conditions.

We have revised this manuscript to include substantial new analyses to address the specific concerns of the reviewer. We wish to emphasize here that we have used similar measures of synaptic efficacy, and experimental conditions that are as well (or in some cases better) controlled, than previous studies.

1) There are several reasons that can explain why the authors failed to see significant differences between sleep and waking; (1) the criteria to define sleep dense and wake dense periods (65% of total time for 4 consecutive hours) are not very restrictive; for instance, in Vyazovskiy et al., 2009 spontaneously awake rats were sacrificed during the dark phase after a long period of continuous waking (1 hour, interrupted by periods of sleep not longer than 4 min), and after spending at least 75 % of the previous 6 hours awake; similar criteria were used in de Vivo et al., 2017 in mice; Liu et al. used a lower cut-off, of 65%, but crucially, the spontaneous waking episodes considered for the study occurred during the first part of the dark phase, when mice and rats are much more active; here, mice were studied in the first part of the light period,

The duration of time, and criteria for defining “sleep dense” and “wake dense” vary substantially across studies that have been used to support sleep and wake-driven changes in synaptic strengths and firing rates. We have included a table with the major such studies, listing these and other experimental details for ease of comparison with our work (Author response Table 1). Most comparable for our mEPSC recordings were the experiments of Liu et al., (2010) which used >65% sleep or wake over 4 hours, and reported differences in mEPSC amplitude/frequency between groups. Other studies with comparable criteria to those used for our mEPSC recordings were Diering et al., (2017) (averaged 69% sleep in 4 hours and 81% wake in 4 hours); Vyazovskiy et al., (2008) (4 hours for fEPSP experiments); and Vyazovskiy et al., (2009) and Watson et al. (2016), which found effects of sleep and wake on firing rate after periods of an hour or less. We note that we include data from wake dense epochs during the dark phase (Figure 3C-G, D. Wake condition).

**Table resptable1:** 

Study Citation	Sleep Definition	Wake Definition	Method for Classification
Liu et al., 2010	>65% 4 hours	>65% 4 hours	Video coding
Vyazovskiy et al., 2008	>75% 6 hours – biochemical experiments Up to 4 hours – Cortical field potential – no explicit threshold	>75% 6 hours – biochemical experiments Up to 4 hours – Cortical field potential – no explicit threshold	EEG and EMG
De Vivo et al., 2017	>75% 7 hours (light period)	>70% 7 hours (dark period) Sleep Dep. 7 hours (light period)	Video coding
Diering et al., 2017	~69% 4 hours (light period) – No Explicit Threshold	~81% 4 hours (dark period) – No Explicit Threshold	Video coding
Watson et al., 2016	Avg. 2/3 hour – No Explicit Threshold	Avg. 1/3 hour – No Explicit Threshold	LFP, EMG, video/motion detector
Miyawaki & Diba, 2016	>30 min with interruptions <1 min	>15 min with interruptions <1 min	LFP/EEG, EMG (3/4 animals)
Vyazovskiy et al., 2009	46/60 min in last hour	50/60 min in last hour Sleep Dep. 4 hours (light period)	EEG, LFP, EMG

While we used a threshold for selection of >65% sleep dense or wake dense for our slice experiments, the average values were higher: 69% for sleep dense, 76% wake for light period wake dense (L. Wake), and 71% for the dark period wake dense animals (D. Wake; Author response image 2; we have added these values to the Results section, lines 159-161). Additionally, we required that the last hour be especially wake or sleep dense (>70% awake/asleep). Furthermore, we had significant variation in wake densities in our dataset (up to 92% wake dense), and we now show that there is no relationship between wake density and mEPSC amplitude across experiments (Figure 3 —figure supplement 1E). As in previous studies we include data on wake dense epochs during the dark phase, and again see no impact of wake on mEPSC amplitudes. Thus our inability to replicate the findings of Liu et al., (2010) cannot be attributed to the duration of our sleep dense and wake dense epochs, but are likely due to other issues with that study, such as lack of information on layer and cell type (as discussed in the Discussion section, lines 389-396, 407412).

Finally, we now show that fEPSPs and evoked spike rates are stable even across the longest sleep and wake dense epochs these animals naturally experience (~5 or 13 hr, respectively; new Figure 6), and this conclusion holds when we use more stringent criteria for sleep and wake dense states (>75% dense, new Figure 6-8). If the longest periods of consolidated sleep and wake these animals experience are not sufficient to drive detectible oscillations in synaptic strength, then the impact of sleep and wake states on synaptic strengths is unlikely to be functionally important.

and as expected and shown in Figure 2B, the probability of single animals to be awake > 65% for 4 consecutive hours was very close to zero; in fact, given figure 2A and B, it is difficult to understand how such long episodes of sustained wake could be found, how many of them, and in how many mice; a figure showing the raw sleep/wake data for all the mice used for the in vitro study (figure 3) should be shown; in line 117, the authors refer to figure 1B, but that figure only shows an example for 60 minutes, not 4 consecutive hours; again, based on figure 2, it is hard to imagine how the authors could find enough episodes in enough mice to perform the experiments; this is further confirmed by the 4 traces shown in figure 3A: none of the 4 examples show >65% waking time for the last consecutive hours; the same issue applies to the layer 2/3 results; also, the legend states that only 3 rats were used for the sleep group, although 4 traces are shown in figure 3A; none of the traces for the inverted wake group are shown, and they should, since results for the inverted waking group actually do show some changes (see below).

We think the reviewer misunderstood what we were plotting in the original figures, and the point we were making about individual variability. We now provide additional data and have revised our figures and description to clarify what we mean. While a given animal has a high probability of experiencing a sleep or wake dense epoch at some point during the day (Figure 1A), when this probability is averaged across animals it can be seen that it fluctuates over ZT but remains pretty low at any given time (new Figure 1B).

This is precisely why circadian time cannot be used as a proxy for sleep dense and wake dense periods, as many studies nonetheless do (discussed in Frank and Cantera, 2014). On the other hand, by following individual animals over several days and quantifying their S-W behavior in real time, we were able to find a sleep or wake dense epoch that met our criteria and ended within a selected circadian window in every animal. Finally, we now show the endpoint analysis for each animal used in the slice experiments in revised Figure 3, Figure 3 – supplement 2, Author response images 2-5 (3-6 animals/condition were used, for a total of 31 animals for the slice experiments). We also illustrate more conventional hypnograms for most animals used in the slice experiments (new Figure 3, Figure 3 – supplement 2; for all hypnograms, Author response images 3-5).

2) There are inconsistencies between the data presented by the authors and their conclusions. Firstly, in Figure 3G the authors did find an increased mEPSC amplitude in inverted wake vs sleep animals, however, they downplayed this point in the text by saying "minor shift"; In fact, it was significant according to K-S test (see Figure 3 G legend). The K-S test but not ANOVA is an appropriate statistical test used for examining the amplitude of mEPSCs, because the distribution of mEPSC amplitude is not normalized and the comparison among means from experimental groups is not appropriate.

We do not believe we are being inconsistent. We see no difference in the *mean* mEPSC amplitude between conditions for any cell type or brain area, and a very small difference in the distribution of mEPSC amplitudes between the Sleep Dense in Light and Wake Dense in Dark in PFC L2/3 (but not in V1). Demonstrating differences in the population mean by cell is the gold-standard measure in the synaptic plasticity field for supporting any contention that there is a change in quantal amplitude between conditions, and whether we use parametric or non-parametric statistics for this comparison none of these differences are statistically significant. Looking at the cumulative distribution of quantal amplitudes by selecting a number of events from each cell can tell you if there are subtle differences in the shape of the amplitude distribution between conditions, but relying on this approach to detect amplitude differences when one does not see it in the cell means is problematic. The KS test is extremely sensitive to very small differences, because it relies on cumulative differences between distributions with many samples (individual mEPSC amplitude or inter-event intervals; we select 60 events per cell to avoid biasing the contribution of any one cell). Further, it is debatable whether one should consider each of these events from a neuron as an independent measurement, as the KS test assumes. Finally, we note that the difference in the median value in the cumulative amplitude distribution in PFC L2/3 is ~5%, much smaller than (for example) the differences reported using anatomical measures (~18%, de Vivo et al., 2017). For all these reasons we consider this to be a minor shift in the amplitude distribution with no significant change in the mean amplitude, which is how we report these data.

Importantly, there is no difference in the cumulative distributions between Wake Dense and Sleep Dense conditions measured during the same circadian period (light period); this modest difference is only detected between conditions at opposite circadian times. We thus interpret this as a possible modest effect of circadian time on the mEPSC amplitude distribution in PFC L2/3 pyramidal neurons. Because we see no such effect in visual cortex it appears to be cell-type specific, as has been reported for other circadian effects on quantal amplitude and synaptic transmission (Bridi et al., 2020). Taken together our data suggest a very modest and cell-type specific effect of circadian time, but not sleep or wake history, on the distribution of quantal amplitudes. We have attempted to clearly lay out our interpretation of these data in the Results section (lines 226-230).

Secondly, In Figure S1F, the authors did show a very significant shift to the left in the cumulative distribution of mEPSC inter-event interval in the WD group as compared with the sleep group in L2/3 PFC neurons (SD-WD p<1e-5), which suggested a higher mEPSC frequency in the WD group than in the sleep group if measured with this parameter. In terms of absolute value, the mEPSC frequency was also higher in WD group than in the sleep group (although it was not significant). Therefore, the statement that synaptic strength was stable across sleep/wake periods is questionable at least in these L2/3 PFC neurons.

Indeed, we reported and discussed differences in mEPSC frequency between conditions in PFC, and we have now moved the mEPSC frequency data to the new Figure 4 and discuss these data more fully. Again there were no significant differences in in the cell averages in any condition, but the cumulative distributions were different in PFC; see our discussion above for relative merits of these two measures. In PFC L2/3 frequency was higher after L. Wake dense then after D. Wake or L. Sleep dense conditions; this suggests that sleep and wake history alone cannot explain this effect.

It is also unclear what these mEPSC frequency differences say about changes in synaptic strength. Many variables can affect spontaneous release frequency without impacting evoked synaptic strength (see revised section of Discussion addressing this, lines 427-438; Choy et al., 2018; Sharma and Vijayaraghavan, 2003; Zhou et al., 2000; Liu and Tsien, 1995). Third, previous studies have arrived at opposite conclusions as to whether sleep deprivation modulates mEPSC frequency in PFC (Liu et al., 2010 found changes; Khlghatyan et al., 2020 did not). Finally, morphological measures in support of SHY document changes in the size but not the number of synaptic contacts (de Vivo et al., 2017), which should correlate with changes in mEPSC amplitude rather than frequency. Thus our hypothesis going into these experiments was that we should see changes in mEPSC amplitude as a function of time spent asleep or awake. We agree this effect on mEPSC frequency is potentially interesting, and we now discuss these points more thoroughly (lines 427-443) – but they do not contradict our statement that all three of our functional measures that are direct correlates of synaptic efficacy (mEPSC amplitude, fEPSP amplitude, and ability to evoke spikes) are unaffected by preceding periods of sleep and wake.

3) It is not clear whether the recording of mEPSCs from naturally wake and sleep rats was well controlled throughout the investigations. It seems that the preparation of slices from these two groups were performed at different times of the day. This means slices were cut at different time points for these two groups (Figure 3A is misleading). Therefore, the variation in slice conditions may mask the difference between groups. Although the authors did sample a big number of cells for each group, I am not sure whether this will help to limit the effect of variation resulting from the slice preparations. Note that Liu et al. took care of running paired experiments, in which one slice from a control animal and one from a waking/sleep deprived animal were always run in parallel the same day, to limit technical variability.

We show the endpoint of all slice experiments in ZT time in the new Figure 3 and Figure 3 —figure supplement 2. The endpoints for all L. Wake dense and L. Sleep dense experiments were within ~4 hours of each other in ZT time and are on average ~8 hours different from the D. Wake dense condition. For the D. Wake condition animals were on an inverted LD cycle so that all slice experiments were performed at a similar time of day for the experimenter. We note also that the mean and variance in our mEPSC amplitude measurements across conditions are quite similar, and are very close to published values from other studies using similar recording and selection criteria (Torrado Pacheco et al., 2020; Lambo and Turrigiano, 2013; Bridi et al., 2020); in contrast, the baseline values in the Liu study were quite variable (for a discussion see Timofeev and Chauvette, 2017). Finally, we do not think that sacrificing animals at precisely the same ZT time when they have had different prior sleep/wake histories (due to individual variability) is a more controlled approach.

4) The rationale for selecting layer 4 should be better justified (line 124); firing rates vary across waking and NREM sleep across the entire thalamocortical system, not just in layer 4; thus, taken alone this is not a compelling reason to select layer 4 neurons; on the other hand, it is well known that after the end of the critical period, the thalamocortical synapses targeting layer 4 of primary somatosensory, auditory, and visual cortex lose most of their ability to undergo plastic changes under physiological conditions, and that ability can be reinstated only by specific manipulations such as prolonged unimodal or crossmodal sensory deprivation or peripheral nerve transection. Thus, it seems that the authors of this study chose to focus on synapses that are known to have little plasticity to test synaptic homeostasis hypothesis, whose main claim is that sleep is the price for plasticity during waking; in fact, if the results related to layer 4 could be trusted (but see all the issues related to selection of behavioral states and minis analysis), then they would actually be a nice confirmation of the main tenet of this hypothesis.

We do not understand the reviewer’s reasoning here. First, SHY posits a global regulation of excitatory synapses, meaning it should be apparent in different brain regions and cell types. We thus chose two brain regions and two distinct excitatory cell types in order to cast a broad net; this included a brain region previously implicated in SHY (Liu et al., 2010). Second, the reviewer is incorrect that we chose a period of time at which thalamocortical synapses onto L4 excitatory neurons are no longer plastic. Our experiments were performed during the classic visual system critical period (which closes after P33) precisely to encompass the period of most pronounced V1 plasticity; we and others have directly demonstrated plasticity of both thalamocortical and intracortical excitatory synapses at this age (see e.g. Miska et al., 2018, Cooke and Bear, 2010; Wang et al., 2013; Kirkwood et al., 1996). We mentioned this in the original version of the manuscript but have now made our rationale more explicit.

Evoked responses:The analysis of the evoked responses is impossible to interpret because too many crucial details and control experiments were not performed.

We have added additional data and analysis to address many of the specific issues raised below, and have completely revised our presentation of these data (see new Figures 5-8).

1) First, the criteria to define prolonged periods of sleep and waking for the evoked responses analysis are not specified and cannot be deduced from Figure 5A, which has no time bar (same problem in Figure 6). In Vyazovskiy et al., a decrease in slope was present only after at least 2 hours of consolidated sleep, or more than one hour of continuous waking (most rats were awake for 2-4 hours). Vyazovskiy also stimulated only twice, before and after sleep or waking, while it seems that in the current study pulses were given every 20 to 40 secs continuously, for days.

In the original manuscript we showed fEPSPs measured across 2 (sleep dense) or 3 (wake dense) hours in state (original Figure 5E); we now include data for much longer periods of time, up to ~5 hours for sleep and ~13 hours for wake dense epochs (new Figure 6-8). The advantage of our approach – continuously sampling fEPSPs at very low frequency during natural periods of waking and sleeping – is that we can follow the kinetics of any changes that might occur with time in a state, and can separately assess the impact of behavioral state, light-dark transitions, and circadian time. Additionally, because we use an optogenetic approach to label a specific set of synapses, we are sampling from a defined and consistent set of synapses across animals. We are now able to show that fEPSP amplitude (Figure 6, 7) and slope (Figure 6 —figure supplement 1) are stable across 65% sleep and wake dense periods that last as long as ~5 hr (SD) and ~13 hr (WD); we see the same stability if we use the more stringent criteria of >75% dense, where we find epochs that extend >2.5 hr for sleep and 8 hr for wake dense. Thus we now examine comparable or longer periods of time, with greater resolution, than in the mentioned study.

2) Second, evoked responses are exquisitely sensitive to neuromodulatory conditions (arousal levels) and subtle changes in arousal could mask any subtle effect due to sleep/waking history. Vyazovskiy et al., took great care in controlling for this factor by delivering the stimuli under a very standardized quiet waking condition, which required 2 investigators watching the animal full time. As they state, "We did not attempt to record evoked responses continuously for several hours in freely behaving rats because it is impossible to maintain the animals in a standard quiet wakefulness for more than a few minutes." Moreover, Vyazovskiy et al., confirmed that changes in slope were present after controlling for response amplitude. Note that their major results were confirmed by comparing high vs low sleep pressure in all 4 behavioral states separately.3) The classifier distinguished 3 states, but not active and quiet waking (line 87). This is a crucial limitation because waking responses vary due to arousal levels (see point 2), and differ between quiet and active waking. There is strong evidence from electrophysiological and calcium imaging data that the activity of V1 neurons is very sensitive to locomotion; thus pooling evoked responses across "waking" is inappropriate.

The reviewer raises an excellent point – we agree that it is important to take behavioral state into account when measuring fEPSPs. Rather than trying to deliver stimuli only in particular states (as in Vyazovskiy above), we instead sampled continuously (at low frequency) while carefully monitoring behavioral state (using LFP and Video); thus we obtained interleaved samples from each state over time. In our original analysis we did not break out fEPSP measurements by the state they were measured in. We agree this is an issue because fEPSP amplitude is rapidly modulated by behavioral state. Furthermore, rodents quickly cycle through different states, for example even a sleep dense epoch is interrupted by short bouts of wake. To address this set of issues we now plot fEPSPs measured within a specific state; for sleep dense epochs we separately plot fEPSPs measured during REM or NREM, and for wake dense epochs we separately plot fEPSPs measured during active and quiet wake. These data are shown in the new figures Figure 6, 7; it can be seen that this does not change our conclusion that fEPSPs remain stable across even very long sleep dense and wake dense epochs.

4) Third, evoked responses are exquisitely sensitive to brain temperature. Very small changes in brain temperature can affect evoked responses and mask any additional effect due to sleep and waking history. Vyazovskiy et al., controlled for this factor by conducting specific experiments in which brain temperature was also measured; in doing so, they could demonstrate that the changes in the slope of the evoked response did not correlate with changes in brain temperature. This issue is especially crucial in the current study, where light pulses were used to evoke the response. On a related matter, the intensity of the laser stimulation should be specified.

Vyazovskiy et al., 2008 showed that there was no significant increase in temperature across an extended waking period, nor did they find that the slope of electrically evoked EPSPs was correlated with brain temperature; thus we do not expect brain temperature to be a confounding variable in our experiments. Since we do not see changes in fEPSP slope with time spent awake or asleep, we do not in any case see how a change in brain temperature could explain our results; one would have to hypothesize that changes in temperature are affecting fEPSPs in such a way as to precisely compensate for a gradual change in fEPSPs due to time spent awake or asleep. The light pulses were 1 ms every 20-40 seconds at a maximum intensity of 18 mW/cm^2^; this duration and intensity of stimulation is not sufficient to change brain temperature (Owen et al., 2019).

Firing rates:The current negative findings relative to firing rates in V1 are at odds with the evidence provided by at least 3 different labs showing that mean firing rates decreases with sleep, including Vyazovskiy et al., in barrel cortex (Nature 2011), Grosmark et al., in the hippocampus (Neuron 2012), Watson et al., in frontal cortex (Neuron 2016), Miyawaki et al., in the hippocampus (Curr Biol 2016, Cell Reports 2019). As for the evoked responses, it is unclear whether the criteria used by Cary and Turrigiano to define prolonged periods of sleep and waking were stringent enough to match those used in other studies. Cary and Turrigiano cite one paper from their lab (line 57) showing that mean firing rates do not change during extended periods of sleep and waking. At the very least, it would be appropriate to quote all the other studies that found the opposite.

We present data on evoked rather than spontaneous firing, but we nonetheless have now included a detailed discussion of the findings from various labs on spontaneous firing rates in different brain areas as a function of time awake or asleep (lines 483-490). We note that the results of the studies cited above do not all agree, and have found different effects (over different timescales) of REM, NREM, and wake on firing rates; this suggests that the impact of sleep and wake on spontaneous firing likely depends critically on brain area, rather than reflecting a global function of sleep and wake states. Our lab now has two separate datasets from V1 where we were able to follow the firing of individual neurons over long periods of time in freely behaving animals as they cycle between many bouts of sleep and wake; we see small differences in firing between states that are expressed rapidly during state transitions, but no significant change across sleep or wake states (Hengen et al., 2016; Torrado Pacheco et al., 2020). We note that rather than measuring ongoing firing (which arises from many internal and external sources) as for these earlier studies, here we probe the ease of evoking spikes using thalamocortical stimulation. Consistent with the other measures of functional synaptic strength we examine, we find that the ability to evoke spikes changes rapidly by state but does not depend on time spent asleep or awake. The criteria for defining prolonged periods of sleep or wake for the evoked firing were the same as for evoked fEPSPs.

The authors state that many units were lost in the course of the several days of recordings. The exact number should be stated.

This number is now included in the methods section (lines 724-726). 93% of cells were well isolated for >1/3 of the full experiment time, and 80% of cells were well isolated for >1/2 of full experiment time. This means that each unit included in our analysis was held across many sleep/wake epochs.

Reviewer #3:This paper asks how states of sleep and wakefulness regulate global synaptic strength at various cortical pyramidal neurons. A major proposition for this question is formulated in the well-known SHY hypothesis, for which there is mostly indirect molecular and structural evidence. Therefore, it is very important to test SHY with direct functional measures of synaptic strength. This paper does so using electrophysiological methods and is, therefore, an important contribution to a long overdue question.The authors depart from a form of homeostatic plasticity that is known to be regulated by sensory experience and largely based on amplitude measurements of mEPSCs. When now applied to sleep and wakefulness, results are overall negative, thus questioning that SHY affects homeostatic plasticity. The experiments are well-done and the results are clear and striking.Still, I would encourage the authors to consider a number of points in a revised version of their manuscript.1) Insufficient information about animal husbandry is provided. All experiments are done in young rats around and shortly after weaning. Weaning changes metabolism and stress levels are high. Synapse growth and development progress rapidly. When were animals weaned relative to the day of surgery? How were they housed prior to and after surgery, and how was recovery from surgery monitored (weight loss and recovery, stress monitoring, etc.)? A time period of 2-3 days for recovery from surgery is very short (~1 week is typical). How much time was given for habituation to the tethering to the recording cables (~1 week is typical)? Is the sleep-wake behavior of the animals stable from 2-3 days after surgery?

We have now added details of our animal husbandry, which followed closely our previously published procedures and timelines (e.g. Hengen et al., 2013, 2016; Torrado Pacheco et al. 2019, 2020). Rats were weaned at P21 and from then were housed with littermates. Surgeries were performed between P22-P26. Animals were housed with littermates during recovery; post-surgical monitoring was approved by the Brandeis IACUC and followed NIH guidelines. All animals received two days of post-operative care comprised of daily injection of Meloxicam and Penicillin. Recovery was rapid and animals were fully recovered (based on normal eating/drinking/weight gain, grooming, and play with littermates) by 2-3 days post-surgery, again consistent with our previous work in these critical period animals. Animals were continuously monitored over several days in their home-cage recording chamber and sleep patterns and fEPSPs generally stabilized within 24 hr of initiation of recording, and prior to data acquisition, as can be seen from our analysis of the ensuing three days of continuous monitoring (Author response image 1).

2) More information on the sleep-wake behavior of these young rodents is also needed. Figure 2 suggests that the typical preference for sleep over wake during the light period found in adult is not there yet. Do these animals show homeostatic regulation of sleep? The SHY hypothesis implies slow-wave activity in the renormalization of synaptic strength. Slow-wave activity is proposed to be key for synaptic scaling during sleep. Therefore, it would be important to show some evidence that slow-wave activity varies with time-of-day and/or after sleep deprivation.

We performed a number of additional analyses of the Sleep/Wake behavior of these young LE rats, and now include these data in the new Figure 1. They indeed have more total sleep in the light phase and more total wake in the dark phase, as reported previously for LE rats of around this age (Frank and Heller, 1997); even in older rats sleep and wake are distributed across both the light and dark phases (Endo et al., 1997; adult WKY rats, Leemburg et al., 2010).

The reviewer raises an excellent set of questions around the amplitude of slow waves and their role in driving plasticity, that we have now dug into. First, we show that our animals exhibit a substantial circadian oscillation in slow wave amplitude, that follows the expected pattern (new Figure 1C). Second, because SHY predicts that synaptic downscaling should be tied to the amplitude of slow waves (i.e. a larger decrease in SWA should correspond to more dramatic downscaling), we looked at the relationship between the change in slow wave amplitude during sleep dense periods and mEPSC amplitude/frequency. We found no relationship, suggesting again that neither mEPSC amplitude or frequency is constitutively downscaled by slow wave sleep.

3) Experiments coincide with critical periods of the visual system. If animals are taken 7-10 days later, after the closure of the critical period, are effects of prior sleep-wake history still negative? What about thalamocortical projections for which the critical period closed much earlier, such as for the whisker-to-barrel system?

The reviewer raises a very interesting set of questions that will be important to examine in future studies. Here we focused on testing some of the core tenets of SHY during a highly plastic period of time when we know we can induce robust synaptic up- and downscaling (Hengen et al., 2016; Torrado Pacheco et al., 2020). We also tested two different brain areas and two different cell types, and used three different measures of functional plasticity. We think our data as they stand provide strong evidence against the idea that sleep serves the global and universal function of constitutively downscaling synaptic strengths. We also acknowledge that this may not be true for all cell types in all brain areas at all developmental times, and now explicitly state this (lines 497-501).

4) Regarding the sleep-wake monitoring and analyses, a major weak point is that scoring of vigilance states is done in 10 sec intervals, which is 2.5x the window commonly used. This means that the duration of NREMS bouts are overestimated because brief arousals will go undetected. There should be an estimation provided for the limited time resolution.

We have now clarified our classification procedures in the revised manuscript. For the real-time sleep/wake classification we used a 10 s interval, to allow us to rapidly determine when a sleep or wake dense epoch had occurred. We subsequently went back and performed a more stringent classification of all sleep/wake behavior with resolution down to 1s. This was sufficient to allow us to detect microarousals and (for example) accurately determine the state in which fEPSPs were elicited. We now add additional detail to the methods (lines 609-611) to clarify this point.

5) The authors only show "Time awake" prior to slice preparation. I suggest they instead show time spent in NREM and REM sleep prior to sacrifice. Possibly, then, animals should be selected based on how much time they spent in NREM only.

We now include additional information on sleep behavior prior to sacrifice in the new Figure 3, Figure 3 —figure supplement 2 (Endpoint state proportions, Author response image 2; all hypnograms, Author response images 3-5). The time spent in NREM during sleep dense epochs is pretty tightly clustered across animals (Author response image 2, Sleep Dense) and is quite different from NREM time in Wake dense epochs during either the L or D phase.

6) Major forms of homeostatic synaptic plasticity in sensory cortices (e.g. effects of monocular deprivation) develop over time scales of days. The authors show themselves in a previous work that after a full day of MD, mEPSC amplitude is reduced to only ~95% of control (Lambo2013). Similarly, in cultures, homeostatic scaling of excitatory or inhibitory synapses is typically observed after 24 h of receptor antagonism. In contrast, here, authors work with time intervals of 4h during which both sleep and wake are present, with mean bout length on the order of hundreds of seconds. This could mean that changes in mEPSC amplitude might simply not be detectable at this point. The mEPSC as a measure for synaptic strength could thus not be sensitive enough. The same considerations might apply for prefrontal cortex, for which the maturational profile is even less known.

The reviewer raises the important question of how sensitive our approach is. We can detect 10-20% mEPSC amplitude changes induced during up- or downscaling, measured ex vivo using the same approach we use here (Lambo and Turrigiano, 2013, Hengen et al., 2013; Torrado Pacheco et al., 2020). Given the variance and sample size, both our mEPSC recordings and fEPSP recordings are capable of detecting an effect size of <10%, which is smaller than that predicted by other studies supporting SHY (Liu et al., 2010; de Vivo et al., 2017). It is possible that sleep and wake constitutively drive such subtle changes in synaptic strength that they are too small to measure over naturally occurring periods of sleep and wake, but then we are not sure what this would mean in terms of the function of these changes. In particular if constitutive changes driven by sleep are too small to induce a detectible functional change in synaptic transmission (as in our evoked spike measurements), then we would suggest they are unlikely to be very important for brain function.

7) More generally, one might wonder about whether mEPSCs are suitable for monitoring changes in synaptic strength. Miniature EPSCs reflect the response of a synapse to the release of a single vesicle. Thalamocortical synapses, however, activate their postsynaptic targets via multivesicular release. This might lead to more vigorous postsynaptic receptor recruitment. Therefore, to fully assess whether or not sleep-wake modify synaptic strength, it would be important to look at how synaptic strength quantified by action-potential-dependent vesicular release is affected. I suggest to use single-fiber stimulation to trigger vesicular release via single action potentials.

We appreciate the reviewer’s point that mEPSCs do not monitor all aspects of synaptic function. We recorded mEPSCs for a number of reasons: (1) this is a classic assay for homeostatic up and downscaling, and we can detect such changes ex vivo when we induce synaptic scaling with visual deprivation or eye reopening; (2) a previous study reported sleep-induced changes in mEPSC amplitude (Liu et al., 2010); and (3) changes in synapse area and in AMPAR accumulation after periods of sleep have been reported recently (Diering et al., 2017; de Vivo et al., 2017), and if these changes are functionally meaningful then they should correlate with changes in mEPSC amplitude. We also used two other functional measures of synaptic efficacy that are sensitive to both pre- and postsynaptic changes in transmission – namely evoked thalamocortical transmission and ability to evoke spikes. These approaches allow us to monitor thalamocortical efficacy in vivo in freely behaving animals, so we think there are major advantages to this over ex vivo measurements of single-fiber efficacy taken at a single point in time. While no measure alone captures all aspects of excitability that might be modulated by sleep and wake, taking these three measures together provides quite a comprehensive survey of synaptic function.

8) Looking at evoked fEPSPs in response to sensory or afferent stimulation has a long tradition but is not a very informative type of data. Evoked fEPSPs are not only composed of sources arising from the synaptic input, but also by the tendency of the network to switch between up and down states. This is particularly the case during nREM sleep because on-going oscillations are strong and excitatory input can switch the networks between states. So, neither amplitude nor slope of these responses, nor their stability across sleep periods, tell much about "synaptic strength".

We respectfully disagree with the reviewer that the amplitude/slope of the thalamocortically evoked fEPSP within L4 is not informative. Many studies have validated that the initial rapid downward deflection arises primarily from activation of synaptic inputs within L4 (Khibnik et al., 2010; Cooke and Bear, 2010), and changes in this amplitude induced by monocular deprivation correlate nicely with changes in thalamocortical EPSCs evoked ex vivo (Miska et al., 2018), using a similar approach to that used here. We have now carefully separated these evoked responses by behavioral state to control for the amplitude differences between states, and regardless of the state in which they are measured we see no sleep or wake-driven changes in fEPSPs. Again, we think each of the approaches we use to probe for functional changes in synaptic strength have some advantages and some disadvantages, but taken together they paint a consistent and compelling picture.

[Editors’ note: what follows is the authors’ response to the second round of review.]

As you will see from the reviewers comments below, the manuscript has been improved but there are remaining issues that need to be addressed, as outlined below. Normally these would require additional experiments, but given pandemic conditions that limit feasibility for such, then at the very least the title, abstract and conclusions need to be moderated to reflect that the conclusions may be limited to an early developmental period. Further, upon careful inspection of the in vivo LFP data, I see that there is a potential confound that needs to be examined. While these are clearly thalamic-dependent LFP responses, as they are recorded in the cortex, and evoked by ChR2 activation of thalamocortical projections, what is not clear is how one might distinguish the specific thalamic fEPSP from the overall LFP response especially with optogenetic stimulation which can lead to a significant fiber volley (see PMID: 27489370).

We thank the editor and reviewers for the thoughtful response to our revised manuscript, and have now made a number of additional changes to address the remaining concerns. The major changes we have made include:

1. We have ensured that the title (line 1), abstract (line 13), and Results section (lines 98108) all emphasize that we examine one developmental period of time, corresponding to the highly plastic visual system critical period. We note that previous studies have spanned a wide age range including earlier, comparable, and later developmental stages.

2. We have performed a number of additional analyses and made textual changes to address the remaining comments of Reviewer 1, detailed below in our response to the points raised.

3. Regarding the contribution of the fiber volley to the fEPSP: the editor is correct that the fiber volley contributes to the early voltage deflections during the fEPSP, whether the responses are evoked by electrical or optical stimulation. We have taken care to minimize the impact of the fiber volley on our estimate of fEPSP magnitude in the following ways. First, we previously characterized the thalamocortical evoked response in great detail in slice recordings from L4 after thalamocortical axons were labeled identically to our protocol here, and found that brief illumination (as we use here) produced a rapid stimulus artifact followed several ms later by a monosynaptic EPSC (Miska et al., 2018 supplemental Figure 1). The initial (predominantly monosynaptic) phase of the in vivo fEPSP we characterize here follows a similar highly stereotyped sequence: there is an immediate stimulus artifact, an early deflection that corresponds to the fiber volley (and is relatively small in L4), and then a rising phase that corresponds temporally to the monosynaptic current we observe in vitro (New Figure 5 – supplement 1). We also note that this optogenetically evoked response looks very similar to previously characterized electrically evoked fEPSPs and visual evoked potentials in L4 (Cooke and Bear, 2010; Niell and Stryker, 2008). Second, our measurement of the rising phase of the fEPSP (slope, measured between 20 and 80% of peak) is thought to be relatively insensitive to contamination by the fiber volley, so is often the standard measure reported for quantifying fEPSP magnitude (Schuman, 1996). This was a reason for measuring and reporting both the fEPSP slope as well as amplitude. Third, the Hass and Glickfield, (2016) study shows that high-frequency optogenetic stimulation in vitro does not evoke a reliable fiber volley, which would greatly complicate interpretation of evoked events.

However, in our study we confined our analysis to very low frequency stimulation (1/201/40 Hz) to avoid this issue. Finally, in addition to measuring the slope, we also measured the first negative peak, which occurs well after the fiber volley (which peaks ~1 ms post stim.; Hass and Glickfield, 2016), but is potentially contaminated by variable polysynaptic events. Notably both measures (slope and amplitude) give the same results for all analyses we performed: namely, that fEPSP varies by behavioral state, but does not change across sleep or wake states. We discuss these points in the Results section (lines 270-282).

Reviewer #1:They authors start with a comprehensive documentation of sleep-wake behavior of young rats during the critical period for vision, followed by an ingenious approach to study synaptic strength as a function of spontaneous recent sleep-wake behaviors, followed by an in-depth analysis of in vitro and in vivo correlates of synaptic strength. I particularly liked the explicit way of the authors in motivating their choice of parameters regarding sleep-wake behavior, of animals during the critical period, and of the synapses studied in vitro or in vivo.The authors carefully conclude that their results do not support the idea that sleep or wakefulness per se lead to global modifications of synaptic strength, at least not in visual and prefrontal cortex. Interestingly, however, the authors identified circadian variations specifically in prefrontal cortex, an observation that is worth pursuing in the future.This manuscript is a long awaited and authoritative approach to challenge the SHY hypothesis with solid experimental quantification of functional synaptic strength.I have a few additional comments to further improve some analysis and their documentation in this study:1) The sleep-wake behavior of these young animals is clearly different than the one from adult animals. It is irregular and polyphasic and there is very little light-dark dependence (see Figure 1B). Therefore, it would be good to show hourly mean times spent in the different vigilance states wake, NREM and REM rather than only the 4h-slidingwindow means.

We now plot hourly means of all states in Figure 1 – supplement 1 (these new plots reveal similar dynamics). However, we do want to stress that our data are actually quite similar to published data from older Long Evans rats in terms of time spent asleep in L vs D, average length of sleep bouts, and other characteristics (Frank and Heller, 1997), and we now make this point clearly in the Discussion section. We also note that there is huge variation in sleep patterns across mammalian species, suggesting that if there is a universal function of sleep it is unlikely to depend critically on the amount or degree of fragmentation.

2) Beyond the mean times, the detailed architecture of sleep-wake behavior in the 4h-windows is also important. Why is this: even if you go for times enriched in wake or sleep, it is not the same whether this enrichment happens in many very brief bouts or in few relatively long bouts. This is particularly the case for sleep, for which fragmentation has a strong effect on plasticity/learning. Therefore, do the animals sleep in consolidated bouts, i.e. what is the mean NREM sleep bout duration? Is this duration variable between Sleep dense and wake dense, and as a function of circadian time?

The mean NREM durations in the SD 4h-window was 161.53 ± 10.4 s, vs. durations of 120.97 ± 7.8 s in the WD window during the same circadian period. Thus sleep bouts were longer during SD periods than WD periods. These values are comparable to those reported previously for Long Evans rats (Frank and Heller, 1997). We now include these values in the Results section, lines 167-168.

3) The word "consolidated" has a strong meaning in the sleep field and as it refers to the mean duration of NREMS bouts – the less they are interrupted by microarousals, the more consolidated NREMS is said to be. It should not be used to describe an enrichment of mean times spent in sleep or wake over a 4-h period (see line 113).

We have changed our language to be more precise (lines 118, 136-137, 140, 178).

4) Line 116: δ power is not the same as the size of slow waves and should not be equated. Slow waves are EEG or LFP graphoelements that at best make up a fraction in the power of the broad δ frequency band used here (0.5-4Hz). Please use these terms carefully and specify the frequency bands upon first use.

We now specify the frequency band upon first use and make clear that we are using δ power as a proxy for slow wave activity (as others have done; Vyazovskiy et al., 2008; Dijk, 2009) (lines 121-123).

5) Line 119. The time-of-day-dependence of δ power is not typically referred to as an oscillation.

We removed the term “oscillation”, lines 124-125

Also, the way the data in Figure 1C analyzed should be checked. It must be done for equivalent amounts of time spent in NREM sleep epochs that are preceded and followed by other NREM sleep epochs. If it is not done like this, the amounts of δ power at different times of day are not weighed equivalently. Therefore, please divide the total time spent in NREMS in the light and in the dark phase into similar amounts and calculate δ power within these bouts. As more time is spent in NREMS in the light phase, subdivisions can be higher in the light than in the dark phase (e.g. 12 time points in the light and 6 in the dark phase). Literature from the Paul Franken lab can be consulted to do this properly.

We redid this analysis using the approach of the Franken lab (1 hr bins during the light period, and 2 hr bins during the dark). The resulting variations in δ power across the L and D cycles look almost identical to our original analysis. We include this plot for the reviewer/editor, and describe this additional analysis in the methods section. We did not include this as a supplemental figure as it seemed redundant, but are happy to include if the editor feels this is important.

6) Lines 168-172: Please explain more quantitatively. What is a long natural wake epoch during the early light phase? What was the criterion to add a novel object? What you do here strictly amounts to a sleep deprivation that should be documented in terms of its effects on quiet vs mobile wakefulness and the increase in the time spent in wakefulness. One could also argue that this is a period of environmental enrichment that has an impact on its own on visual plasticity. These animals might also show a greater increase in δ power upon the end of wakefulness due to greater amounts of sleep loss. These caveats should be quantified whenever possible and discussed.

We have now added additional details to the methods section to document how we extended wakefulness (lines 606-612). In brief, we waited until animals had experienced ~50% wakedense in the previous 4 hr, and then encouraged further wakefulness by moving, removing, or adding new toys or stirring the bedding; this procedure was generally initiated 1.25-2.25 hr before slicing, and maintained until animals had reached criterion for wake density. It is important to note that all animals experience the removal and addition of new toys and changes to bedding regularly, so although the frequency of these manipulations is higher during this wake extension they are familiar procedures to the animals. In general, our environment is enriched compared to the standard home cage environment. We sacrificed animals immediately after wake encouragement without allowing them to sleep, so cannot say whether they would have experienced higher δ power upon entering NREM. That said, based on our analysis of the full circadian cycle in these and other conspecific animals, we predict the δ power would be higher.

7) Figure 3 —figure supplement 1. This figure indicates somewhat worrisomly that the increase in δ power at light onset compared to the end of the light period is extremely variable – from somewhere between >40%, which is very high, to ~17%, which is very low. In addition, the example recording shown in panel A is close to 50%, so where is that datapoint represented in panel B? What happened with these animals during the darkphase that their sleep pressure at light onset is so variable? Overall, I am not sure that this analysis is particularly helpful because it relates an endpoint measure of synaptic amplitudes to an unknown starting point measure.

The differences in δ power drop during the light phase between animals reflects the fact that individual animals show a lot of variability in when they sleep – as we document carefully in this study (e.g. Figure 1). This is an important take-home message of our study. We would note that individual variability is generally not carefully examined in studies of rodent sleep, so it is difficult to compare our data to previous studies. In the example in A, average δ power is ~30% (not 50%) when averaged over the time window indicated.

Overall, I am not sure that this analysis is particularly helpful because it relates an endpoint measure of synaptic amplitudes to an unknown starting point measure.

One of the predictions of SHY is that slow waves drive synaptic downscaling, and in turn downscaling should reduce slow waves (and thus δ power) (Tononi and Cirelli, 2014; Tononi, 2009). That was our rationale for comparing the *change* in δ power across the preceding sleep period for each animal, with the synaptic strengths at the end of the sleep period. We also included this analysis to ensure that a change in synaptic strength did not become apparent in animals that experienced a large drop in δ power. For these reasons we think it is worth keeping this analysis as a supplemental figure.

8) Line 624. What are slice behavioral data?

We have clarified our language here (lines 638-639)

9) The finding that evoked field potential amplitudes in visual cortex were larger in NREMS than in wakefulness during both the light and the dark phase is intriguing and in contrast with previous observations on evoked auditory field responses (see e.g. the literature from Yaniv Sela et al.,). Rather what has been seen is that the secondary outward components of the evoked responses are disproportionately increased during NREM sleep.

In Sela et al., 2020, they find that auditory stimuli produce comparable firing rate responses between states in auditory cortex and somewhat diminished responses in perirhinal cortex. They further find that a subset of neurons termed “late-responding” (>40 ms post stimulus) in auditory cortex show a reduction in response during NREM. Larger cortical evoked potentials in NREM have been found by others using optogenetic stimulation (Matsumoto et al., 2020), or auditory stimuli (Hall and Borbely, 1970; see also Nir et al., 2015 for a discussion of the mixed results on this topic). We find that all aspects of the response (amplitude, slope, second outward component) are larger in NREM as compared to wake, in accord with the majority of studies in cortex.

Evoked field responses are, as already mentioned in my first review, also problematic because they can be contaminated because of on-going oscillatory activity. Can the authors better describe when these responses were elicited in response to on-going up- and downstates, for example? And how these components were removed to isolate the evoked field response?

We have addressed potential variability in fEPSPs during NREM in three ways. First, we calculated the coefficient of variation of responses in all 4 vigilance states, and CV was not larger in NREM than in other states (NREM ~0.3; other states ~0.3-0.4). This suggests that slow waves do not much impact our fEPSP measurements. Second, we measured fEPSPs separately in REM (where there are no up and down states) and NREM for the same sleep states, and in neither case do we see a relationship between time spent asleep and fEPSP slope or amplitude (Figure 6B, C). And third, we used a “sandwich” approach to detect changes across sleep states, where we compared fEPSPs in wake states before and after a prolonged period of sleep. These later two analyses do not depend on measurements during NREM and yield the same result.

Reviewer #2:In the revised manuscript, the authors included important and interesting new analyses, but no new experiments have been undertaken, and the key issue remains that the data set, limited to results obtained in juvenile animals and analyzed in some novel, yet untested ways, is not ideal for reaching strong conclusions.My main points I raised previously remain the same:1. The experiments have been performed in juvenile animals, which prevents direct comparisons with most previous studies. I recommend that the title of the manuscript should state the age of animals to avoid misunderstanding.

We have modified the title, abstract, and conclusions to emphasize that we examine one developmental period of time, corresponding to the highly plastic visual system critical period. We note that previous studies have spanned a wide age range including earlier, comparable, and later developmental stages to that used here.

2. The experiments were performed too soon after a major invasive surgical procedure, which is certainly expected to affect sleep and wake quality, because the immune system and thermoregulation can be still compromised especially in young animals.3. The overall wake-sleep pattern is too fragmented for a straightforward comparison between effects of wake and sleep, and I still do not see evidence provided that homeostatic sleep pressure is increased during time periods referred to as "wake-dense" epochs, and decreases during "sleep-dense" epochs.4. The chronic fEPSP experiment shows massive state- and light-dark-dependent instantaneous variations in the amplitude of responses, which makes it very difficult to extract any meaningful sleep-wake history dependent changes, and the potential direct influence of arousal, behaviour and brain temperature on evoked responses remains not addressed.

We addressed each of these points very carefully in the previous revisions to the manuscript. Briefly, we showed that sleep/wake behavior is fully recovered prior to the initiation of experiments. Other aspects of animal health (grooming, weight gain, play with siblings, and measures of activity in V1) are also back to baseline. The sleep wake patterns of these juvenile LE rats are approaching adult patterns (Frank and Heller, 1997), and evidence for SHY has been provided using much younger animals with very immature sleep patterns (de Vivo et al., 2019).

Finally, we have used a variety of approaches to analyze our fEPSP data that are independent of the rapid state-dependent changes in amplitude, and they provide a consistent picture.

[Editors’ note: what follows is the authors’ response to the second round of review.]

Reviewer #1:The authors did an excellent job replying to my concerns. In my view, the study now makes abundantly lear that it studies critical period synapses. The study also has appropriately addressed remaining analytical concerns, with one exception:This concerns the re-analysis of the data presented in the Reviewer Figure 1. I appreciate that the authors re-binned their data according to the light-and dark phase. However, this analysis is NOT about simple binning. It is about calculating δ power for similar times spent in NREMsleep. Therefore, it is about dividing the total amount of time spent in NREMS in different percentiles in the light and dark phase. Accordingly, the datapoints will not be placed in regular time intervals, but they will be displaced according to where the mean values of the time bin come to lie.While I can understand that this can be considered as a detail, I encourage the authors to present their main figure according to the standards in the field.

We understand the reviewer’s point. We went back through publications from the Franken lab (as suggested by the reviewer) and adopted their approach, so that δ is calculated across the same amount of NREM sleep. The results are very similar to both of our previous methods of calculating the drop in δ power. We have now replaced Figure 1C with the new version of this figure, and modified the methods section accordingly.

My second remaining concern is that the authors should once again read the recommendations of the Senior Editor to use careful wording regarding the evidence against SHY given all that was discussed. For example, I still see the last sentence of the abstract "…strong evidence against the view that sleep drives widespread downscaling" unchanged.

Upon reflection, we think this final phrase in the abstract is unnecessary and have simply deleted it. We have considered our wording throughout and have modified our phrasing in places to avoid overstating our case (for example in the final sentence of the introduction).

Finally, the Supplement to Figure 5 seems not necessary to me.

We agree and have removed this figure.

Reviewer #2:The change in the title is welcome, but no attempt has been made to address my other comments. My original comments were:

We made numerous changes to the manuscript in the previous round of revisions to address the concerns of the reviewer. On several major points we disagree with the reviewer and outline our reasons below.

– The experiments were performed too soon after a major invasive surgical procedure, which is certainly expected to affect sleep and wake quality, because the immune system and thermoregulation can be still compromised especially in young animals.The current version of the manuscript states, for example, "Animals were given 2-3 days of recovery before data collection, at which point sleep/wake behavior had stabilized and was comparable to unsurgerized animals." No data were provided to support this statement, and in my view this is unacceptable for any study that attempts to address physiological functions of sleep.

We used a protocol for these chronic recordings that we have established and published on extensively, comprising several independent datasets all of which show that animals have recovered both behaviorally and by electrophysiological measures prior to the start of recordings (Hengen et al., 2013, 2016; Torrado Pacheco et al., 2019, 2021). Animals are habituated to the recording chamber on the third day post-surgery and are fully recovered at this point (eating and drinking normally, gaining weight, grooming normally, etc.; following NIH guidelines and our IACUC approved protocol). This timeframe gives us the greatest recording stability for these continuous chronic recordings. In addition to showing that activity in V1 is stable by this point (citations above), we show here that evoked responses are stable over the several days of recordings used in our analysis (Figure 5D). Finally, we verified that sleep patterns in our animals have also stabilized prior to the initiation of experiments; these data were provided as Reviewer Figure 1 in our original re-submission and we now include them as Figure 1 supplement 2. Further, our data on sleep/wake distributions look very similar to previously published data from Long Evans rats of comparable ages (e.g. Frank and Heller, 1997; Frank et al., 2017). Taking all of this together, we do not think the length of recovery after surgery is a confounding factor in our data.

The overall wake-sleep pattern is too fragmented for a straightforward comparison between effects of wake and sleep, and I still do not see evidence provided that homeostatic sleep pressure is increased during time periods referred to as "wake-dense" epochs, and decreases during "sleep-dense" epochs.

As we pointed out previously, the degree of sleep fragmentation we see is exactly what is expected in rodents of this age. We can find and analyze long periods of sleep and wake that are as “dense” as those in the literature in support of SHY, and yet see no sign of constitutive downscaling. We do provide evidence that sleep pressure (measured in the usual way, as a change in δ power, Figure 1C) indeed oscillates as expected across periods of sleep and wake. Finally, we point out that we can readily detect the gating of homeostatic plasticity by sleep and wake in animals of the same age (Hengen et al., 2016; Torrado Pacheco et al., 2021).

I would like to refer the authors to extensive literature which describes modelling of sleep homeostasis (based on δ power, or firing rates), specifically studies that show individual examples of how Process S changes as a function of sleep-wake states. From those studies it is clear that the dynamics occuring as a function of sleep-wake states are expected to be relatively slow, and one cannot expect prominent changes during wake or sleep states unless these are consolidated.

The reviewer does not acknowledge that in our revised manuscript we show that even periods of time as long as 5 hours (sleep) and 12 hours (wake) do not drive detectible changes in synaptic strength. If these periods of time are not sufficient, then constitutive sleep and wake-dependent synaptic changes will simply not be detectable in rodents (or in many other animals that do not exhibit long consolidated sleep/wake states). It is also worth noting that previous experiments in support of SHY have claimed to see changes driven by as little as 4 hr of sleep or wake in rodents of comparable ages.

– The chronic fEPSP experiment shows massive state- and light-dark-dependent instantaneous variations in the amplitude of responses, which makes it very difficult to extract any meaningful sleep-wake history dependent changes, and the potential direct influence of arousal, behaviour and brain temperature on evoked responses remains not addressed.

The modulation of evoked transmission by vigilance state is an interesting and expected phenomenon, and one cannot measure plasticity in vivo without taking it into account. Our experimental design in fact allowed us to deal rigorously with these state-dependent neuromodulatory changes, by allowing us to track responses across sleep and wake and independently analyze those that were measured within a given state (NREM or REM for sleep, active or quiet wake for wake). Finally, we used several independent analytic approaches that all gave the same answer, and we found the same results whether we measured mEPSCs ex vivo, or evoked fEPSPs and spikes in vivo. Each measure has potential downsides but we feel that together they provide coherent and compelling evidence for stability of these measures across sleep and wake states.